# HARA: A Unified Framework for Hardware-Efficient Non-Linearity in Transformers

## Abstract

The deployment of modern Transformer models on edge devices is critically bottle-necked by computationally intensive non-linear operators like GELU, Softmax, and LayerNorm, which demand diverse and power-hungry specialized hardware units. Existing functional approximation techniques suffer from two critical failures: they are function-specific, leading to hardware bloat, and they rely on unstable heuristics that yield poor accuracy. We introduce HARA (Hybrid Arithmetic-ReLU Networks Approximation), a framework that resolves these issues by systematically replacing all such operators with a single, canonical architecture built from simple arithmetic primitives and a shallow ReLU network. HARA's core algorithmic innovation is an Optimized Parameter Initialization pipeline that employs dynamic programming to systematically derive near-optimal parameters, ensuring high-fidelity approximation and robustness where heuristic methods fail. Crucially, hardware synthesis estimations project that HARA's unified approach reduces the silicon area for non-linear processing by over 60% compared to using separate, specialized functional units. We demonstrate across four modern architectures (BERT, Swin, LLaMA, and Stable Diffusion) that these significant hardware savings are achieved with negligible impact on model performance (e.g., $< 0.1\%$ accuracy change) and are fully compatible with 8-bit quantization. By systematically co-designing software approximations with a simplified hardware target, HARA provides a practical and extensible paradigm for deploying state-of-the-art Transformer models on resource-constrained devices.

## 1 Introduction

Transformers have revolutionized machine learning, yet their deployment on resource-constrained edge devices is hindered by high computational complexity. A significant portion of this complexity arises not from matrix multiplications alone, but from non-linear operators like activation functions (GELU, SiLU, Tanh, Sigmoid), Softmax, and normalization layers (LayerNorm, RMSNorm). These functions rely on computationally intensive operations such as exponentials, divisions, and square roots, which necessitate specialized, power-hungry functional units in hardware, increasing silicon footprint and energy consumption.

To tackle this efficiency challenge, two primary strategies have emerged: quantization and functional approximation. Quantization reduces the numerical precision (e.g., from FP32 to INT8) of existing operations, primarily accelerating arithmetic-heavy operations like matrix multiplication. However, it does not alter the fundamental operations themselves; an 8-bit quantized model still requires the hardware to compute $exp$, $sqrt$, and $div$. In contrast, functional approximation aims to replace these complex operations with simpler, hardware-friendly alternatives. While using ReLU networks for this purpose is well-established (Zhang et al., 2024a; Liu & Chen, 2024; Kim et al., 2023b; Chen et al., 2019), existing methods are ill-suited for creating a practical, unified hardware solution. They suffer from two critical limitations: 1) Function-Specific Designs, where bespoke approximation techniques are used for each operator (e.g., GELU vs. Softmax) (Zhang et al., 2024a; Vasyltsov & Chang, 2021; Yu et al., 2022). This fragmentation requires different specialized hardware circuits for each function, defeating the goal of a simple, unified accelerator. and leading to increased silicon area and design complexity 2) Suboptimal and Heuristic Parameterization, where heuristic or direct-training methods yield poor accuracy and, critically, fail to generalize across different input ranges—a catastrophic failure for real-world deployment that we demonstrate in our experiments.

In this work, we present HARA, a unified framework that systematically resolves these issues. HARA replaces key non-linear operators with a canonical ReLU-arithmetic computation pattern, addressing the aforementioned limitations with two core innovations: a **unified architecture** to eliminate hardware bloat and a **principled optimization pipeline** to ensure robust, high-fidelity approximation. Our contributions are:

1. **A Unified Operator Framework** that decomposes diverse non-linear functions into simple arithmetic operations $(+, -, \times, \ll)$ and a shallow (1-layer) ReLU network, enabling massive hardware resource sharing.

2. **An Optimized Parameter Initialization Pipeline** that leverages dynamic programming to systematically find near-optimal parameters for the ReLU approximator using operator's symmetry, achieving significantly higher accuracy than direct training methods.

3. **Comprehensive Validation and Hardware Analysis** demonstrating that HARA-approximated models (BERT, Swin, LLaMA, Stable Diffusion) maintain performance ($< 0.1\%$ change) and are robust to post-training 8-bit quantization, while hardware estimations project over a 60% reduction in silicon area for non-linear processing units compared to a baseline with specialized hardware.

## 2 RELATED WORK

This section reviews the theoretical foundations and practical applications of function approximation in neural networks, identifying the critical gaps in existing methods that motivated the development of HARA.

**Theoretical Foundations of Function Approximation**

The use of neural networks for function approximation is theoretically well-grounded. Seminal works Cybenko (1989); Hornik et al. (1989) proved that single-hidden-layer networks can uniformly approximate any continuous function on compact subsets of $\mathbb{R}^n$. More recent studies Yarotsky (2017); He et al. (2018) have advanced this theory by characterizing the optimal approximation rates for ReLU networks and establishing the critical depth-width trade-offs required to efficiently represent complex functions. While these works confirm the expressive power of ReLU networks, they primarily focus on theoretical bounds rather than providing practical, systematic methods for approximating the specific, diverse operators found in modern Transformers.

**Practical Approximation Methods in Transformers**

Applying these theories in practice presents distinct challenges across different operator classes in Transformer architectures. **Activation Functions:** Functions like GELU (Dauphin et al., 2017) offer smoother gating mechanisms than ReLU but are more computationally expensive. **Softmax:** The exponential terms in the Softmax function are a primary bottleneck for hardware implementation, leading to prior work (Vasyltsov & Chang, 2021; Yu et al., 2022) focused on replacing them with piecewise linear approximations. **Normalization:** Techniques like LayerNorm (Ba et al., 2016) introduce variance-sensitive non-linearity through square root and division operations, which are challenging to implement efficiently. To address these challenges, several hardware-aware approximation frameworks have been proposed. Notably, NN-LUT (Yu et al., 2022) utilizes a neural network to generate hardware-friendly lookup tables for non-linear operations. Similarly, RI-LUT (Kim et al., 2023a) explores an alternative reconfigurable integer-based LUT design for efficient hardware implementation. These methods validate the general approach of replacing complex functions with simpler, specialized hardware primitives.

**Optimization Methods for Function Approximation**

Optimization techniques are crucial for balancing the competing demands of computational efficiency and model accuracy in function approximation. Foundational research He et al. (2015) into deep residual networks highlighted the importance of principled initialization strategies for improving the training dynamics of ReLU-based architectures, providing valuable guidance for any ReLU-based approximation task. While subsequent theoretical work (Ramachandran et al., 2017; Yarotsky, 2018)established bounds on the network complexity required to achieve a given approximation error, these studies focused primarily on expressive power rather than on developing practical,

actionable optimization strategies. Other efficiency-focused research (Bhalgat et al., 2020) has introduced learnable quantization techniques that indirectly relate to this problem, but these methods are primarily concerned with weight quantization and do not fundamentally replace the non-linear operators themselves. Collectively, while prior work provides theoretical justification and explores related efficiency concepts, it lacks a systematic and practical optimization pipeline specifically designed for the robust, high-fidelity approximation of diverse non-linear operators within a unified framework.

**The Research Gap: The Need for a Unified and Systematic Framework**

Despite this progress, existing methods suffer from two fundamental limitations that hinder the development of a truly general and efficient hardware accelerator.

First, they remain **functionally fragmented**. Most approaches employ operator-specific designs—a bespoke solution for GELU, another for Softmax, and so on. This fragmentation leads to hardware bloat, as it requires a collection of different specialized circuits, defeating the goal of a simple, area-efficient, and unified accelerator. Second, their parameterization is often **heuristic and suboptimal**. Many methods rely on direct, unconstrained training to find the parameters for their approximators. As our experiments in Section 4.2.1 demonstrate, such naive approaches are unstable, yield lower accuracy, and fail to generalize across the wide input ranges encountered in real-world scenarios. Consequently, a critical gap exists for a framework that is both unified across all operators to maximize hardware resource sharing and systematic in its parameter optimization to guarantee high-fidelity, robust performance. HARA is designed to explicitly fill this gap by providing a single, canonical architecture and a principled, DP-based optimization pipeline.

## 3 HARA FRAMEWORK METHODOLOGY

HARA provides a comprehensive and systematic framework for replacing computationally expensive non-linear operators in Transformer models with a unified, hardware-efficient arithmetic-ReLU architecture. Our methodology is built upon three integrated components: (1) a single, canonical architecture based on a hardware-software co-design paradigm, (2) a mathematically rigorous, multi-stage optimization pipeline to derive its parameters, and (3) a strategic decomposition of complex operators into hardware-friendly primitives. This approach systematically adapts the software to run efficiently on simpler, unified hardware, maximizing resource sharing and efficiency.

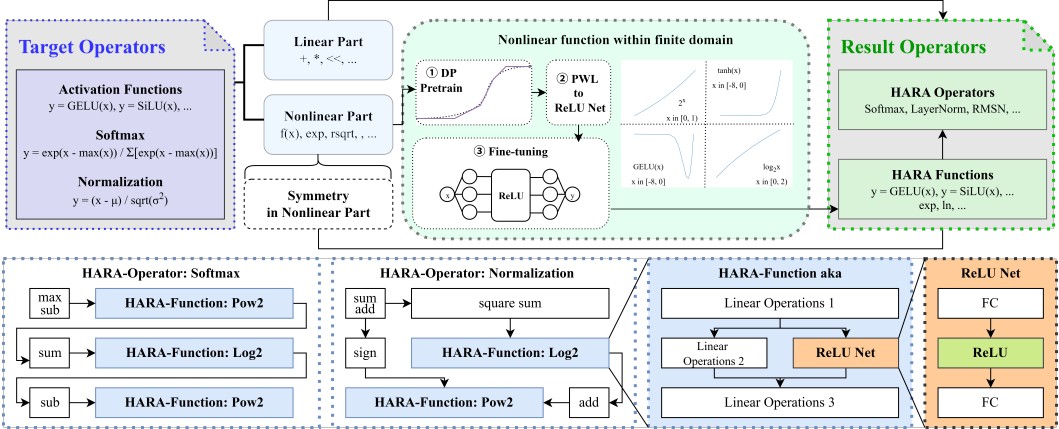

Figure 1: An overview diagram of HARA framework

### 3.1 THE UNIFIED HARA ARCHITECTURE: A HARDWARE-SOFTWARE CO-DESIGN PARADIGM

The cornerstone of the HARA framework is its hardware-software co-design paradigm. Instead of designing distinct hardware for each function, HARA maps all targeted non-linear operators onto a single, consistent software architecture. This architecture's core is a shallow, single-hidden-layer

Rectified Linear Unit (ReLU) network, whose canonical formulation is given by:

$$f(x) = W_2(\text{ReLU}(W_1 x + b_1)) + b_2 \tag{1}$$

This unified software model enables HARA's primary architectural benefit: replacing multiple, power-hungry specialized hardware units for operations like $exp$, $sqrt$, and $div$ with a single, reconfigurable hardware block known as the Unified ReLU Network (URN). This strategy drastically reduces hardware complexity and silicon footprint. Hardware synthesis estimations project that this unified approach reduces the silicon area for non-linear processing by over 60% and yields power savings of over 51% compared to a baseline (BL) with separate, specialized units. The basic unit of proposed hardware is the URN block, which is composed of configurable look-up tables (CLUTs), and auxiliary functions (AFs, used as additional arithmetic and control logics). HARA is consisted of several parallel URN blocks, sum generator (SG), max block (MB), local buffer (LB) and one controller. By pre-loading function-specific parameters into the CLUTs, the same URN can be dynamically reconfigured to process any required non-linear operator, maximizing throughput and hardware utilization.

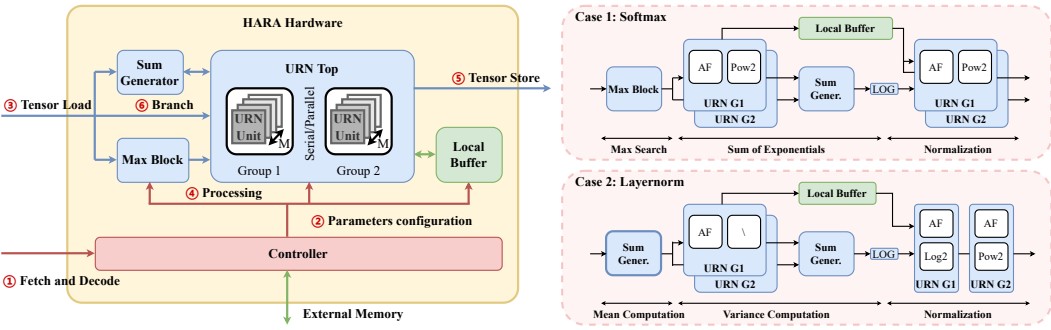

Figure 2: HARA hardware architecture

## 3.2 OPTIMIZED PARAMETER INITIALIZATION

To make this unified architecture effective, the parameters ($W_1$, $b_1$, $W_2$, $b_2$) must be derived with high precision. A naive attempt to train the ReLU network directly often leads to suboptimal, unstable solutions that fail to generalize, (as shown in Figure 3). To overcome this, HARA employs a principled, three-stage pipeline to initialize the network parameters, ensuring a high-fidelity and robust approximation.

1. **Stage 1: Optimal Breakpoint Selection via Dynamic Programming.** The process begins by framing the task as a Piecewise Linear (PWL) approximation problem. Given a target function, we employ a dynamic programming (DP) algorithm to identify the optimal break-point locations that globally minimize the mean squared error (MSE). This step is crucial as it yields an optimal PWL representation, avoiding the pitfalls of unstable heuristic methods.

2. **Stage 2: Analytical Conversion to ReLU Network Parameters.** With the optimal slopes $k$ and intercepts $B$ of the PWL function determined, we leverage the analytical relationship between a PWL function and a single-layer ReLU network to directly compute the initial weights and biases. The full derivation of this inverse mapping is detailed in Appendix A.1 (as derived in Equations 7–9). A key constraint, $k[0] = 0$, is enforced to ensure the approximation correctly models the required asymptotic behavior (i.e., $f(x) \to 0$ as $x \to -\infty$), which also makes the mathematical conversion well-posed and robust.

3. **Stage 3: Fine-Tuning.** The parameters derived from the analytical conversion provide a near-optimal starting point. A final, brief fine-tuning stage using the Adam optimizer is then applied to further minimize any residual approximation error, polishing the result to achieve maximum fidelity.

---

**Algorithm 1** DP-based Initialization of HARA Parameters

---

1: **Input:** Data vectors $x, y$; Number of segments $N$
2: **Output:** Network parameter vectors $n, m, B$
3: $i_{\text{breaks}} \leftarrow \text{DynamicProgramming}(x, y, N)$ ▷ Find indices of $N-1$ optimal breakpoints
4: $i \leftarrow \text{Concat}(0, i_{\text{breaks}}, \text{length}(x) - 1)$ ▷ Form $N+1$ indices for $N$ segments
5: $k \leftarrow \text{zeros}(N+1), b \leftarrow \text{zeros}(N+1)$ ▷ Enforce $f(x) \to 0$ as $x \to -\infty$
6: **for** $j \leftarrow 0$ to $N-1$ **do**
7: $\quad x_0 \leftarrow x_{i_j}, \quad x_1 \leftarrow x_{i_{j+1}}$
8: $\quad y_0 \leftarrow y_{i_j}, \quad y_1 \leftarrow y_{i_{j+1}}$
9: $\quad k_{j+1} \leftarrow (y_1 - y_0)/(x_1 - x_0)$ ▷ Calculate slope of segment $j$
10: $\quad b_{j+1} \leftarrow y_1 - k_{j+1} \cdot x_1$ ▷ Calculate bias of segment $j$
11: **end for**
12: $n_j \leftarrow k_j - k_{j-1}$, for $j = 1, \ldots, N$ ▷ Calculate first-layer weights
13: $m_j \leftarrow \text{sign}(n_j)$, for $j = 1, \ldots, N$ ▷ Calculate second-layer weights
14: $(m_B)_j \leftarrow b_j - b_{j-1}$, for $j = 1, \ldots, N$
15: $B_j \leftarrow (m_B)_j/m_j$, for $j = 1, \ldots, N$ ▷ Calculate first-layer biases
16: **return** $n, m, B$

---

In Algorithm 1, $x$ is discretized input domain of the target nonlinear function and $y$ is corresponding target function values at each input point in $x$. $N$ is the desired number of linear segments for the piecewise linear approximation. $i$ is a vector containing the indices of the $N-1$ optimal breakpoints, augmented with the start (0) and end indices of the domain $x$. $k$ denotes a vector of slopes for the $N+1$ piecewise linear segments. It is initialized with $k_0 = 0$ to satisfy the asymptotic boundary condition $f(x) \to 0$ as $x \to -\infty$. $b$ is a vector of bias terms for the $N+1$ linear segments in the PWL approximation. $n$ is the calculated weights for the first layer of the equivalent two-layer ReLU network. $m$ is the calculated weights for the second layer of the network, constrained to be either $+1$ or $-1$. $B$ is the calculated biases for the first layer of the ReLU network.

## 3.3 APPLICATION TO TRANSFORMER OPERATORS

This systematic pipeline is applied to the full spectrum of non-linear operators found in modern Transformers by categorizing them based on their mathematical properties.

### 3.3.1 ACTIVATION FUNCTIONS (GELU, SiLU)

A key challenge with activation functions is that they are defined over an infinite domain, whereas ReLU networks are most effective at approximating functions over finite intervals $[a, b]$. While such networks can achieve high accuracy within the training interval, approximation quality outside this range critically depends on the slopes of the terminal segments, as illustrated in Figure 3. HARA addresses this by exploiting the inherent symmetry and asymptotic properties of these functions, as summarized in Table 1. Target activation functions can be expressed by approximation functions concentrated on the negative domain. $g(x)$ in Table 1 represents the approximation function that closely matches the original function for $x < 0$, while maintaining that $g'(-\infty) = 0$. Consequently, the approximation problem is transformed from finite domain to infinite domain. For example, functions like GELU and SiLU can be decomposed into a linear part ($ReLU(x)$) and a non-linear part that is even and decays to zero.

Table 1: Symmetry and asymptotic properties of common activation functions

| AF | Formula | Symmetry | $f'(-\infty)$ | $f(-\infty)$ | Negative Approx |
|---|---|---|---|---|---|
| Sigmoid | $\frac{1}{1+e^{-x}}$ | $f(x) - c$ is odd | 0 | 0 | $g_{\text{Sigmoid}}(-|x|) + c$ |
| Tanh | $\frac{e^x - e^{-x}}{e^x + e^{-x}}$ | odd | 0 | -1 | $g_{\text{tanh}}(-|x|)$ |
| GELU | $x \cdot \Phi(x)$ | $f(x) - \text{ReLU}(x)$ is even | 0 | 0 | $g_{\text{GELU}}(-|x|) + \text{ReLU}(x)$ |
| SiLU | $\frac{x}{1+e^{-x}}$ | $f(x) - \text{ReLU}(x)$ is even | 0 | 0 | $g_{\text{SiLU}}(-|x|) + \text{ReLU}(x)$ |
| Softplus | $\ln(1 + e^x)$ | $f(x) - \text{ReLU}(x)$ is even | 0 | 0 | $g_{\text{Softplus}}(-|x|) + \text{ReLU}(x)$ |

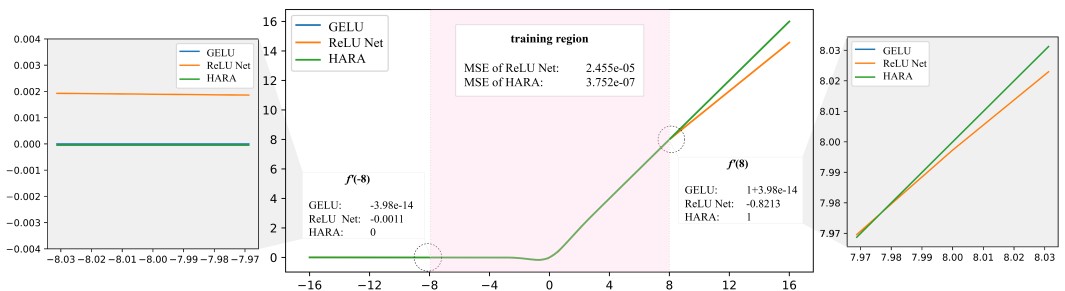

Figure 3: HARA vs conventional ReLU approximation

### 3.3.2 COMPLEX OPERATORS (SOFTMAX AND NORMALIZATION)

Operators like Softmax and LayerNorm are challenging due to their reliance on hardware-intensive exponentials, divisions, and square roots. Instead of approximating these complex functions directly, HARA first transforms their computational structure into a series of simpler operations built around two core mathematical primitives: power of 2 ($2^x$) and logarithm base 2 ($log_2 x$). The mathematical details of these decompositions are provided in Appendix A.2. The resulting structures are as follows:

$$\text{Softmax}(x) = \frac{\exp(\hat{x})}{\sum \exp(\hat{x})} = \exp\left(\hat{x} - \log \sum \exp(\hat{x})\right) = 2^{\left(\hat{x} \cdot \log_2 e - \log_2 \sum 2^{\hat{x} \cdot \log_2 e}\right)} \quad (2)$$

$$\text{LayerNorm}(x) = \frac{x - \mu}{\sqrt{\sigma^2}} = \text{sgn}(\bar{x}) \cdot 2^{\left(\frac{1}{2}\log_2 M + \log_2 |\bar{x}| - \frac{1}{2}\log_2 \sum_{j=1}^M \bar{x}^2\right)} \quad (3)$$

In which: $\hat{x} = x - \max(x)$, $\mu = \frac{1}{M}\sum_{j=1}^M x_j$, $\sigma^2 = \frac{1}{M}\sum_{i=1}^M (x_i - \mu)^2$, $\bar{x} = Mx - \sum_{j=1}^M x_j$, $M$ is the sequence length of $x$.

All complex non-linearities are thus isolated into the Pow2 and Log2 functions. We then use the optimized initialization pipeline to create highly accurate ReLU network approximators for these primitives over their required finite domains (e.g., [0, 1) for Pow2 and [1, 2) for Log2), which illustrated in Appendix A.2. The full Softmax or LayerNorm operation is then reconstructed by chaining these HARA functions with simple arithmetic operations, completely eliminating the need for expensive, specialized hardware units.

## 4 EXPERIMENTS AND RESULTS

We conducted a comprehensive set of experiments to validate the HARA framework across four critical dimensions: (1) the fundamental accuracy and robustness of our approximation algorithm compared to existing methods, (2) the projected hardware efficiency gains from our unified architecture, and (3) the end-to-end performance impact on state-of-the-art Transformer models, including compatibility with quantization. Detailed results, including additional tables and hardware analysis, are provided in Appendix A.3.

### 4.1 EXPERIMENTAL SETUP

To ensure a thorough evaluation, we selected three categories of nonlinear operations and four representative Transformer architectures spanning diverse domains, as listed in Table 2.

**1. BERT-base (Devlin et al., 2019) (BERT)** for natural language understanding (SQUAD v2.0 dataset (Rajpurkar et al., 2018)),

**2. Swin-Tiny (Liu et al., 2021) (Swin)** for computer vision (ImageNet-1k (Deng et al., 2009)),

**3. LLaMA3.2-3B (Grattafiori et al., 2024) (LLaMA)** for language generation (WikiText-2 (Merity et al., 2016)), and

**4. Stable Diffusion 3.5 Medium (Esser et al., 2024) (DiT)** for text-to-image synthesis (SDCI dataset (Zhang et al., 2024b)).

Performance was measured using standard metrics for each task: Exact Match (EM) and F1 Score for BERT, Top-1/Top-5 accuracy for Swin, perplexity (PPL) for LLaMA, and Human Preference Score (HPSv2) for DiT. Our implementation uses PyTorch and replaces all target non-linear operators (GELU, Softmax, LayerNorm, etc.) with their HARA counterparts without any architectural modifications.

Table 2: Nonlinear operations in transformer models

| Models | GELU | Sigmoid | SiLU | Tanh | Softplus | Softmax | LayerNorm | RMSNorm |
|--------|------|---------|------|------|----------|---------|-----------|---------|
| BERT   | √    |         |      |      |          | √       | √         |         |
| Swin   | √    |         |      |      |          | √       | √         |         |
| LLaMA  |      |         | √    |      |          | √       |           | √       |
| DiT    | √    |         | √    |      |          | √       | √         | √       |

## 4.2 ALGORITHMIC SUPERIORITY AND HARDWARE EFFICIENCY

Before evaluating full models, we first establish the core advantages of HARA's methodology at the operator level.

### 4.2.1 SUPERIOR APPROXIMATION ACCURACY

We compared HARA's approximation accuracy against representative function approximation baselines, NN-LUT and RI-LUT. As shown in Table 3, HARA achieves a Mean Squared Error (MSE) that is several orders of magnitude lower than these directly trained methods across all tested operators (GELU, Softmax, LayerNorm). More importantly, HARA demonstrates superior robustness; as the complexity (hidden dimension, a.k.a HD) increases, its approximation error consistently and predictably decreases, while the error for the baseline methods stagnates or behaves erratically. This validates the stability and effectiveness of our systematic optimization approach.

Table 3: HARA vs NNLUT vs RILUT cross GELU/Softmax/LayerNorm approximation (MSE)

|    | GELU | | | Softmax | | | LayerNorm | | |
|----|-------|-------|----------|----------|----------|----------|-----------|----------|----------|
| HD | NNLUT | RILUT | HARA | NNLUT | RILUT | HARA | NNLUT | RILUT | HARA |
| 2  | 2.07e-03 | 8.13e-05 | **2.36e-05** | 1.34e-06 | 2.90e-09 | **2.32e-10** | 1.32e-01 | 2.79e-02 | **4.17e-06** |
| 4  | 7.77e-04 | 4.78e-05 | **3.49e-06** | 4.14e-07 | 1.80e-11 | **1.43e-11** | 2.79e-01 | 7.39e-04 | **7.04e-07** |
| 8  | 8.08e-06 | 4.53e-05 | **3.74e-07** | 3.26e-07 | 2.35e-12 | **5.08e-13** | 2.30e-01 | 7.30e-04 | **1.24e-07** |
| 16 | 2.07e-06 | 4.48e-05 | **3.20e-08** | 7.88e-08 | 3.70e-13 | **1.14e-14** | 2.22e-02 | 3.86e-05 | **2.27e-08** |

### 4.2.2 ABLATION STUDY: THE CRITICAL ROLE OF DP-BASED INITIALIZATION

To isolate the source of HARA's accuracy, we performed an ablation study on our initialization pipeline. The results in Table 4 demonstrate that a "Naive" direct training approach yields high approximation errors. In contrast, using Dynamic Programming ("DP") to find an optimal PWL representation first reduces the MSE by several orders of magnitude. The final fine-tuning stage ("DP w/ FT") further refines the parameters, consistently achieving the lowest error across all functions. This study confirms that our principled, DP-based initialization is the key driver of HARA's superior performance, not merely the network architecture itself.

### 4.2.3 PROJECTED HARDWARE EFFICIENCY GAINS

The primary motivation for HARA is to reduce hardware complexity by replacing multiple specialized non-linear units with a single, unified architecture. To quantify this benefit, we performed synthesis

Table 4: Ablation study on parameter initialization stages

| Methods | GELU | Sigmoid | SiLU | Tanh | Softplus | Softmax | LayerNorm | RMSNorm |
|---------|------|---------|------|------|----------|---------|-----------|---------|
| Naive | 1.38e-03 | 2.91e-03 | 2.74e-04 | 7.56e-04 | 5.44e-02 | 1.13e-09 | 9.85e-06 | 5.80e-06 |
| DP | 1.34e-06 | 1.89e-06 | 3.19e-06 | 4.31e-06 | 2.52e-05 | 2.49e-12 | 6.91e-08 | 1.43e-07 |
| DP w/ FT | **1.89e-07** | **5.07e-07** | **5.14e-07** | **1.03e-06** | **4.77e-06** | **2.88e-13** | **5.74e-08** | **6.06e-08** |

estimations using a 6nm cell library to compare a baseline design (using separate, specialized LUT-based units for Softmax, LayerNorm, and GELU) against our single and basic core block of unified HARA implementation (URN).

Table 5: Comparison of estimated area and power costs for different operators/units

| Operator/Unit | Implementation Method | Est. Area Cost($um^2$) | Est. Power Cost($mW$) |
|---------------|----------------------|------------------------|------------------------|
| BL Specialized Units | | | |
| Softmax | Log(LUT)/Div(LUT) | 6890.24 (91AU) | 0.346 (61PU) |
| Layernorm | Sqrt(LUT)/Div(LUT) | 6816.94 (90AU) | 0.432 (77PU) |
| GELU | Polynomial Approx.(LUT) | 6349.44 (84AU) | 0.387 (69PU) |
| Total Baseline | Multiple Specialized Units | 20056.62 (265AU) | 1.165 (207PU) |
| HARA(HD=8) | URN | **7560.85** (100AU) | **0.563** (100PU) |
| Estimated Savings | Unified vs. Separate | 62.3% | 51.7% |

As shown in Table 5, HARA's single, reconfigurable URN is projected to perform all non-linear functions using only $7,560um^2$ of silicon area, a reduction of over 62% compared to the $20,056um^2$ required by the specialized units. This unification also yields an estimated power saving of over 51%. These significant hardware savings are the direct result of our co-design approach.

### 4.3 END-TO-END MODEL PERFORMANCE AND QUANTIZATION ROBUSTNESS ANALYSIS

Having established HARA's algorithmic and architectural advantages, we evaluated its impact on end-to-end model performance, demonstrating that these hardware savings are achieved with a negligible toll on accuracy.

Table 6: HARA approximation with optimal dimension and quantization across transformer models

| | BERT | | Swin | | LLaMA | DiT |
|---|------|-----|------|-----|-------|-----|
| | EM | F1 | Top1_Acc | Top5_Acc | PPL | HPSv2 |
| Baseline | **80.038** | **87.616** | **81.182** | 95.516 | **7.814** | 0.2724 |
| HARA (8,8,8) | 80.02↓ | 87.615↓ | 81.170↓ | **95.538↑** | 7.819↓ | **0.2731↑** |

Table 6 summarizes the performance of the four models after replacing all non-linear operators with their HARA-approximated versions, using an efficient configuration (hidden dimension 8) and standard 8-bit post-training quantization. The results show an exceptional preservation of model performance. For BERT, F1 score remains virtually unchanged ($87.616 \rightarrow 87.615$), with a negligible EM drop of 0.013. For Swin, Top-1 accuracy decreases around 0.01, and Top-5 accuracy slightly improves (95.538 vs. 95.516). For LLaMA, there is only a marginal increase in perplexity (7.819 vs. 7.814). For DiT, the human preference score is nearly identical (0.2731 vs. 0.2724). These results convincingly demonstrate that HARA functions as a drop-in replacement for computationally expensive operations, maintaining model accuracy within 0.1% of the baseline across all tasks. Furthermore, its compatibility with 8-bit quantization confirms that HARA's hardware-friendly architecture is a practical and robust solution for deploying state-of-the-art Transformer models on resource-constrained edge devices.

## 5 DISCUSSION AND LIMITATIONS

Our results demonstrate that HARA can replace heterogeneous non-linear operators in Transformers with a unified, hardware-friendly architecture while maintaining model accuracy. The core contribution is an algorithmic and mathematical foundation that enables a paradigm shift in hardware-software co-design: instead of building complex hardware to fit the software, we adapt the software to run efficiently on simpler, unified hardware. The success of our DP-based initialization, which dramatically outperforms direct training baselines, suggests that principled, systematic optimization is crucial for high-fidelity function approximation. This automated approach also ensures extensibility; as new activation functions emerge, they can be incorporated by finding new parameters for the existing HARA architecture, without requiring a bespoke hardware redesign.

**Limitations and Future Work.** The primary limitation of this study is that our hardware benefits are based on synthesis estimations rather than a full physical implementation and post-layout analysis. While our estimations in Section 4.2.3 project significant area and power savings , a full ASIC synthesis would be required to obtain definitive measurements of latency and performance on a physical chip. However, the viability of deploying simplified hardware for non-linear operations is strongly supported by prior work. Frameworks such as NN-LUT and RI-LUT have already demonstrated the feasibility of implementing efficient, LUT-based hardware for function approximation. HARA's contribution is complementary and foundational: we provide a superior algorithmic method for generating the parameters for such hardware. As our direct comparison in Table 4 shows, HARA's systematic DP-based optimization yields approximations that are orders of magnitude more accurate and robust than the direct-training methods often used by such frameworks.

Therefore, we posit that our software-level validation provides a critical and necessary prerequisite for the development of a next-generation unified accelerator. Future work should focus on the physical hardware synthesis to quantify these real-world efficiency gains. Another promising direction is extending HARA into the training loop, which could potentially allow models to learn entirely new, hardware-optimal non-linearities from scratch.

## 6 CONCLUSION

The deployment of modern Transformer models on resource-constrained edge devices is critically bottlenecked by a diverse set of computationally intensive non-linear operators that demand specialized and power-hungry hardware. This paper introduced HARA (Hybrid Arithmetic-ReLU Networks Approximation), a unified framework that resolves this challenge through a systematic hardware-software co-design approach. At its core, HARA replaces heterogeneous operators like GELU, Softmax, and LayerNorm with a single, canonical arithmetic-ReLU architecture, enabling massive hardware resource sharing. The success of this unified model is driven by our key algorithmic innovation: an Optimized Parameter Initialization pipeline that uses dynamic programming to systematically derive near-optimal parameters. This principled method ensures high-fidelity approximation and robustness where unstable heuristic or direct-training methods fail. Our comprehensive validation across four modern architectures—BERT, Swin, LLaMA, and Stable Diffusion—demonstrates that this unification is achieved with negligible impact on model performance, typically a change of less than 0.1% in key metrics. Furthermore, the HARA framework is fully compatible with 8-bit quantization, a critical requirement for edge deployment. These minimal performance trade-offs are coupled with significant hardware savings, as synthesis estimations project a reduction in silicon area for non-linear processing by over 60% compared to using separate, specialized functional units. By co-designing software approximations with a simplified hardware target, HARA provides a practical and extensible paradigm for deploying state-of-the-art AI. It effectively bridges the gap between the advanced capabilities of large Transformer models and the stringent constraints of edge computing, establishing a new path forward for efficient and powerful AI on any device.

ETHICS STATEMENT

This work adheres to the ICLR Code of Ethics. The research presented herein is focused on the development of computationally efficient methods for neural networks, specifically by reducing the hardware footprint and power requirements of non-linear operators in Transformer models. A primary positive societal impact of this research is the potential reduction in the environmental and economic costs associated with deploying large-scale AI models, thereby promoting more sustainable and accessible technology.

In this study, no human subjects or animal experimentation or the collection of new data was involved. All experiments were performed on established, publicly available academic datasets (SQUAD v2.0, ImageNet-1k, WikiText-2, and the SDCI dataset), and we have complied with all their respective usage licenses. We have taken care to avoid any biases or discriminatory outcomes in our research process. No personally identifiable information was used, and no experiments were conducted that could raise privacy or security concerns. We are committed to maintaining transparency and integrity throughout the research process.

REPRODUCIBILITY STATEMENT

We have made every effort to ensure that the results presented in this paper are reproducible. The experimental setup, including model configurations, choice of hidden dimensions, and post-training quantization parameters, is described with sufficient detail in the paper to allow for complete replication. The hardware synthesis estimations reported in Table 5 were conducted using a standard 6nm cell library; we will provide detailed documentation of the synthesis methodology to ensure our hardware efficiency claims can be independently verified. All datasets used are public and widely available, with sources cited in the manuscript. We are confident these resources will enable the research community to fully reproduce and build upon our findings.

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

## USE OF LARGE LANGUAGE MODELS (LLMS) STATEMENT

Large Language Models (LLMs) were used to aid in the writing and polishing of the manuscript. Specifically, we used an LLM to assist in refining the language, improving readability, and ensuring clarity in various sections of the paper. The model helped with tasks such as sentence rephrasing, grammar checking, and enhancing the overall flow of the text.

It is important to note that the LLM was not involved in the ideation, research methodology, or experimental design. All research concepts, ideas, and analyses were developed and conducted by the authors. The contributions of the LLM were solely focused on improving the linguistic quality of the paper, with no involvement in the scientific content or data analysis.

The authors take full responsibility for the content of the manuscript, including any text generated or polished by the LLM. We have ensured that the LLM-generated text adheres to ethical guidelines and does not contribute to plagiarism or scientific misconduct.

## A  APPENDIX

### A.1  MATHEMATICAL FORMULATION OF PWL TO RELU NETWORK CONVERSION

This section details the analytical process for converting the parameters of a Piecewise Linear (PWL) function into the weights and biases of an equivalent single-hidden-layer ReLU network. This conversion is the mathematical foundation for the second stage of our Optimized Parameter Initialization pipeline.

As shown in (Yu et al., 2022) and summarized in Equation 4 and 5, a ReLU network defines a PWL function. Our goal is to solve the inverse problem. Given a set of optimal slopes $k[i]$ and breakpoints derived from our dynamic programming stage, we need to find the network parameters $W_1$, $b_1$, $W_2$, $b_2$.

$$\sum_{i=1}^{N} W_2[i] \cdot \text{ReLU}(W_1[i] \cdot x + b_1[i]) + b_2 \Rightarrow \begin{cases} k[0] \cdot x + B[0], & x \in (-\infty, p[0]] \\ k[1] \cdot x + B[1], & x \in (p[0], p[1]] \\ \vdots \\ k[N] \cdot x + B[N], & x \in (p[N-1], \infty) \end{cases} \quad (4)$$

$$p[i] = -\frac{b_1[i]}{W_1[i]}, \quad k[i] = \sum_{j=1}^{i} W_2[j] \cdot (W_1^+[j], b_1^+[j]), \quad B[i] = \sum_{j=i+1}^{N} W_2[j] \cdot (W_1^-[j], b_1^-[j]) \quad (5)$$

where the positive and negative part of $W_1$ and $b_1$ are defined as Equation 6.

$$W_1^+[i], b_1^+[i] = \begin{cases} W_1[i], b_1[i], & \text{if } W_1[i] \geq 0 \\ 0, 0, & \text{if } W_1[i] < 0 \end{cases}, W_1^-[i], b_1^-[i] = \begin{cases} 0, 0, & \text{if } W_1[i] \geq 0 \\ W_1[i], b_1[i], & \text{if } W_1[i] < 0 \end{cases} \quad (6)$$

The formulation of piecewise linear (PWL) functions is examined through the construction of ReLU networks. $N = 3$ is chosen as an example for illustration. The relationship between the segment slopes $k[i]$ and the network weights is shown in Equation 7. A direct solution for the network parameters from these equations is ill-posed, as the system is underdetermined.

$$\begin{aligned} k[0] &= W_2[1] \cdot W_1^-[1] + W_2[2] \cdot W_1^-[2] + W_2[3] \cdot W_1^-[3] \\ k[1] &= W_2[1] \cdot W_1^+[1] + W_2[2] \cdot W_1^-[2] + W_2[3] \cdot W_1^-[3] \\ k[2] &= W_2[1] \cdot W_1^+[1] + W_2[2] \cdot W_1^+[2] + W_2[3] \cdot W_1^-[3] \\ k[3] &= W_2[1] \cdot W_1^+[1] + W_2[2] \cdot W_1^+[2] + W_2[3] \cdot W_1^+[3] \end{aligned} \quad (7)$$

It is helpful to reverse above construction process by deriving the parameters of the ReLU network from PWL representation. To facilitate this derivation, we introduce auxiliary variables $s_i$ and $A_i$ as:

$$s_i = \begin{cases} 0, & W_1[i] \geq 0 \\ 1, & W_1[i] < 0 \end{cases}, \quad A_i = W_2[i] \cdot |W_1[i]| \quad (8)$$

Then the slopes can be rewritten compactly as:

$$\begin{aligned} k[0] &= -(s_1 A_1 + s_2 A_2 + s_3 A_3) \\ k[1] &= -(s_1 A_1 + s_2 A_2 + s_3 A_3) + A_1 \\ k[2] &= -(s_1 A_1 + s_2 A_2 + s_3 A_3) + A_1 + A_2 \\ k[3] &= -(s_1 A_1 + s_2 A_2 + s_3 A_3) + A_1 + A_2 + A_3 \end{aligned} \quad (9)$$

While $A$ can be readily solved by taking differences between adjacent equations, the solution for $s$ is underdetermined due to rank deficiency. However, imposing the constraint $W_1 > 0$, which leads

to $k[0] = 0$, renders the system well-posed. The condition $W_1 > 0$ simultaneously achieves two goals: (1) ensuring that the ReLU network asymptotically approaches $y = 0$ as $x \to -\infty$, consistent with the behavior summarized in Table 1; and (2) enabling a well-defined inverse mapping from a given set of PWL parameters to corresponding ReLU network parameters. With this constraint, $A$ can be resolved, and then obtain $W_1 = |A|$, $W_2 = \text{sign}(A)$. From the relation $p[i] = \frac{-b_1[i]}{W_1[i]}$, $b_1$ is subsequently determined. The procedure completes the reconstruction of a ReLU network that faithfully recovers a PWL function initialized at $y = 0$. Consequently, This robust, analytical conversion is a key reason HARA avoids the instability of direct-training methods.

## A.2 DECOMPOSITION OF COMPLEX OPERATORS INTO PRIMITIVES

To handle complex operators like Softmax and LayerNorm, HARA first decomposes them into a series of simple arithmetic operations and two core non-linear primitives: power of 2 ($2^x$) and logarithm base 2 ($log_2(x)$). This section details the mathematical basis for these decompositions.

Pow2 Decomposition: As shown in Equation 10, any input x can be decomposed into its integer ($\lfloor x \rfloor$) and fractional ($x - \lfloor x \rfloor$) parts. The $2^{\lfloor x \rfloor}$ term is implemented efficiently in hardware as a simple bit-shift ($\ll$) operation. Therefore, the ReLU network is only required to approximate the function $f(x) = 2^x$ over the finite domain $[0, 1)$, significantly simplifying the approximation task.

$$2^x = 2^{\lfloor x \rfloor} \cdot 2^{x - \lfloor x \rfloor} = 2^{x - \lfloor x \rfloor} \ll \lfloor x \rfloor \tag{10}$$

Log2 Decomposition: The decomposition for the logarithm leverages the IEEE 754 floating-point representation of a number, as shown in Equation 11. The exponent term ($E_x - 127$) can be extracted directly. The non-linear component is isolated to the mantissa, which is normalized to the finite domain $[1, 2)$. The ReLU network is thus only tasked with approximating $f(x) = log_2(x)$ over this small, fixed interval.

$$\log_2 x = \log_2 \left( 1 + \frac{M_x}{2^{23}} \right) + (E_x - 127) \tag{11}$$

To align with activation function approximation pipeline mentioned in Section 3.1, we apply simple transformations to Pow2 and Log2 functions. Equation 12 shows the final transformations applied to these primitives to ensure they satisfy the $f(x) \to 0$ as $x \to -\infty$ condition required by our initialization pipeline.

$$f_{\text{train\_pow2}}(x) = \begin{cases} 0, & x \in (-\infty, 0] \\ 2^x - 1, & x \in (0, 1] \end{cases} \quad f_{\text{train\_log2}}(x) = \begin{cases} 0, & x \in (-\infty, 1] \\ \log_2 x, & x \in (1, 2] \end{cases} \tag{12}$$

Using the same approximation pipeline described in Section 3.1, we obtain ReLU networks for Log2 and Pow2, which are then assembled into complete Softmax and normalization computations according to Equations 2 and 3.

## A.3 DETAILED EXPERIMENTAL RESULTS AND HARDWARE ANALYSIS

This section provides supplementary data and analysis supporting the results presented in Section 4. We include a detailed breakdown of approximation accuracy, hardware synthesis estimations, and end-to-end model performance across different configurations.

### A.3.1 APPROXIMATION ACCURACY ANALYSIS

Table 7 contains the raw Mean Squared Error (MSE) values for each HARA-approximated non-linear operator at different hidden dimensions (HD). The data shows a clear and consistent trend: as the hidden dimension of the ReLU network increases, the approximation error decreases significantly, typically by an order of magnitude with each doubling of HD. This demonstrates the systematic accuracy-complexity trade-off enabled by the HARA framework. Notably, for complex operators like Softmax and LayerNorm, the MSE reaches exceptionally low values (e.g., $2.08e - 08$ for LayerNorm at HD=16), validating the effectiveness of our decomposition strategy.

Table 7: MSE between original and approximated operations.

| HD | GELU | Sigmoid | SiLU | Tanh | Softplus | Softmax | LayerNorm | RMSNorm |
|----|------|---------|------|------|----------|---------|-----------|---------|
| 2 | 2.36e-05 | 4.44e-05 | 8.14e-04 | 1.99e-04 | 1.18e-04 | 1.19e-10 | 4.10e-05 | 3.26e-06 |
| 4 | 3.49e-06 | 4.75e-06 | 9.70e-05 | 2.07e-05 | 1.31e-05 | 2.01e-11 | 6.22e-06 | 4.27e-06 |
| 8 | 3.74e-07 | 4.33e-07 | 1.08e-05 | 1.72e-06 | 1.13e-06 | 1.15e-12 | 4.29e-07 | 3.39e-07 |
| 16 | 3.20e-08 | 4.62e-08 | 2.07e-06 | 1.27e-07 | 8.73e-08 | 4.73e-14 | 2.08e-08 | 5.32e-10 |

### A.3.2 HARDWARE EFFICIENCY ANALYSIS

This subsection details the hardware cost-benefit analysis of the HARA architecture.

Figure 4 visualizes the direct trade-off between hardware area cost and approximation accuracy (MSE) for the GELU function. Increasing the hidden dimension provides lower error at the cost of a near-linear increase in silicon area, allowing hardware designers to make principled choices based on their specific accuracy and budget constraints.

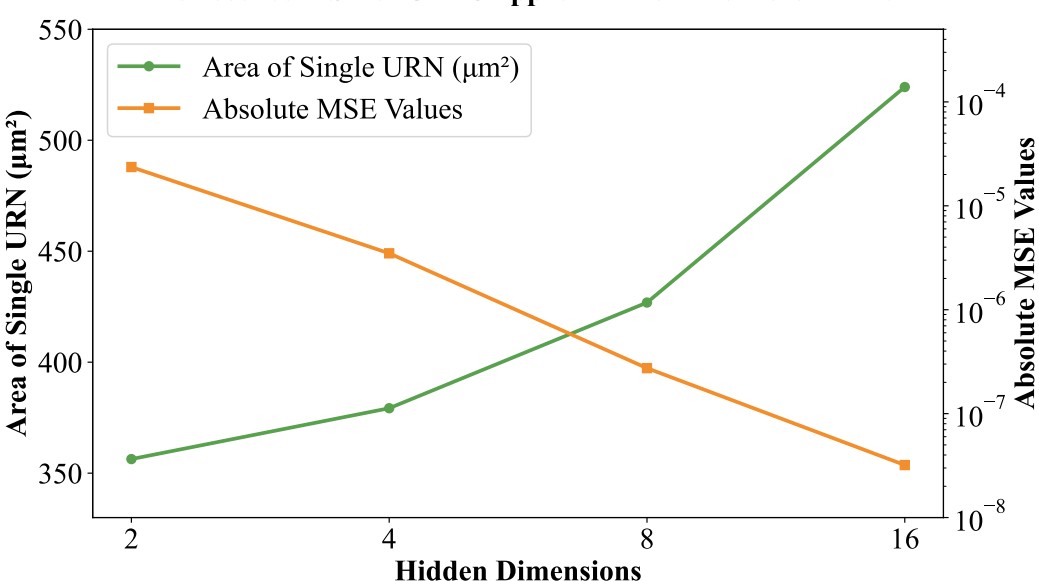

Figure 4: The relations between area cost and MSE of HARA at different HDs

Table 8 provides a comparative breakdown of the area and power costs of a single HARA URN (with HD=8) versus other LUT-based approximation methods like NN-LUT and RI-LUT. The results highlight that the HARA URN is the most area-efficient design.

Table 9 lists the estimated latency, power, and area for the unified HARA hardware which consists of 16 URN modules when configured to execute the three primary operator types. Note that the area cost is constant ($7560.85um^2$), as the same physical URN is reconfigured for each function, which is the core benefit of our unified approach.

### A.3.3 END-TO-END MODEL PERFORMANCE ANALYSIS

Tables 10 through 16 provide a comprehensive breakdown of the end-to-end model performance, supplementing the summary in Table 6.

Tables 10-13 show the impact on model performance (BERT, Swin, LLaMA, DiT) when replacing non-linear operators with HARA approximators of varying hidden dimensions (2, 4, 8, 16). The results consistently show that while low-HD approximators (HD=2, 4) can cause a performance drop,

Table 8: Hardware area/power comparison

| Components | Linear-LUT (256-entries) | NN-LUT (16-entries) | RI-LUT (16-entries) | HARA(URN) (8-entries) |
|---|---|---|---|---|
| Factorization | —— | —— | 47.83 | 47.83 |
| LUT Ctrl | 508.32 | 36.64 | 36.64 | 24.74 |
| LUT Memory | 915.82(SRAM) | 193.83(FF) | 193.83(FF) | 100.93(FF) |
| FP16 MADD | 94.57 | 94.57 | 94.57 | —— |
| INT MADD | —— | —— | —— | 43.56 |
| Scale | —— | —— | 58.82 | 58.82 |
| Auxiliary Function | 150.99 | 150.99 | 150.99 | 150.99 |
| Total Area($um^2$) | 1669.7 | 476.03 | 582.68 | 426.87 |
| Relative Area | 3.911 | 1.12 | 1.37 | 1 |
| Total Power($mW$) | 0.0533 | 0.0294 | 0.0428 | 0.0362 |
| Relative Power | 1.47 | 0.81 | 1.18 | 1 |

Table 9: PPA results of typical nonlinear operations in HARA

| Operator | CLUT Configuration | Relevant modules | Latency($cycles$) | Power($mW$) | Area($um^2$) |
|---|---|---|---|---|---|
| LayerNorm | POW2, LOG2 | URN, SG, LB | SL/8+15 | 0.495 | 7560.85 |
| Softmax | POW2 | URN, MB, LB | SL/8+15 | 0.637 | 7560.85 |
| GELU | GELU | URN | 5 | 0.557 | 7560.85 |

particularly for activation functions, performance is almost perfectly restored at HD=8 and HD=16, matching the baseline.

Table 10: BERT validation on different HDs with each nonlinear operation approximation

| HD | GELU | | Softmax | | LayerNorm | | All | |
|---|---|---|---|---|---|---|---|---|
| | EM | F1 | EM | F1 | EM | F1 | EM | F1 |
| BL | 80.038 | 87.616 | 80.038 | 87.616 | 80.038 | 87.616 | 80.038 | 87.616 |
| 2 | 65.866 | 76.710 | 79.972 | 87.585 | 79.981 | 87.577 | 65.487 | 76.406 |
| 4 | 79.934 | 87.586 | 80.028 | 87.606 | 80.057 | 87.637 | 79.934 | 87.570 |
| 8 | 80.028 | 87.626 | 80.038 | 87.616 | 80.009 | 87.603 | **80.028** | **87.624** |
| 16 | 79.991 | 87.588 | 80.038 | 87.616 | 80.038 | 87.616 | 79.962 | 87.579 |

Tables 14-16 demonstrate the robustness of HARA-approximated models to post-training quantization. The results confirm that HARA's approximations are fully compatible with 8-bit quantization ([8,8]), maintaining performance very close to the baseline across all models.

Table 11: Swin validation on different HDs with each nonlinear operation approximation

| HD | GELU | | Softmax | | LayerNorm | | All | |
|---|---|---|---|---|---|---|---|---|
| | Top1_A | Top5_A | Top1_A | Top5_A | Top1_A | Top5_A | Top1_A | Top5_A |
| BL | 81.182 | 95.516 | 81.182 | 95.516 | 81.182 | 95.516 | 81.182 | 95.516 |
| 2 | 71.972 | 90.272 | 81.150 | 95.524 | 81.180 | 95.502 | 71.866 | 90.162 |
| 4 | 81.086 | 95.500 | 81.186 | 95.520 | 81.196 | 95.522 | 81.074 | 95.504 |
| 8 | 81.184 | 95.514 | 81.182 | 95.516 | 81.184 | 95.524 | **81.184** | 95.512 |
| 16 | 81.176 | 95.520 | 81.182 | 95.516 | 81.184 | 95.520 | 81.176 | **95.520** |

Table 12: LLaMA validation on different HDs with each nonlinear operation approximation

| HD | GELU | Softmax | RMSNorm | All |
|---|---|---|---|---|
| | PPL | PPL | PPL | PPL |
| BL | 7.814 | 7.814 | 7.814 | 7.814 |
| 2 | 9.802 | 7.814 | 7.820 | 9.818 |
| 4 | 7.917 | 7.814 | 7.815 | 7.921 |
| 8 | 7.817 | 7.814 | 7.814 | 7.818 |
| 16 | 7.813 | 7.814 | 7.814 | **7.814** |

Table 13: DiT validation on different HDs with each nonlinear operation approximation

| HD | GELU | SiLU | Softmax | LayerNorm | RMSNorm | All |
|---|---|---|---|---|---|---|
| | HPSv2 | HPSv2 | HPSv2 | HPSv2 | HPSv2 | HPSv2 |
| BL | 0.2724 | 0.2724 | 0.2724 | 0.2724 | 0.2724 | 0.2724 |
| 2 | 0.2551 | 0.2017 | 0.2725 | 0.2724 | 0.2731 | 0.1891 |
| 4 | 0.2740 | 0.1760 | 0.2725 | 0.2722 | 0.2725 | 0.1824 |
| 8 | 0.2728 | 0.2711 | 0.2725 | 0.2724 | 0.2724 | 0.2694 |
| 16 | 0.2725 | 0.2723 | 0.2725 | 0.2726 | 0.2722 | **0.2731** |

Table 14: BERT validation on various qunatized precision with each nonlinear operation approximation

| BW | GELU | | Softmax | | LayerNorm | |
|---|---|---|---|---|---|---|
| | EM | F1 | EM | F1 | EM | F1 |
| BL | 80.038 | 87.616 | 80.038 | 87.616 | 80.038 | 87.616 |
| FP16 | 80.028 | 87.626 | 80.038 | 87.616 | 80.009 | 87.603 |
| [4,4] | 79.603 | 87.356 | 79.612 | 87.379 | 78.912 | 86.638 |
| [8,4] | 79.64 | 87.625 | 79.697 | 87.597 | 78.732 | 87.589 |
| [8,8] | 80.019 | 87.356 | 80.028 | 87.386 | 79.981 | 86.615 |
| [16,4] | 79.631 | 87.336 | 79.688 | 87.349 | 78.761 | 86.768 |

Table 15: Swin validation on various qunatized precision with each nonlinear operation approximation

| BW | GELU | | Softmax | | LayerNorm | |
|---|---|---|---|---|---|---|
| | Top1_A | Top5_A | Top1_A | Top5_A | Top1_A | Top5_A |
| BL | 81.182 | 95.516 | 81.182 | 95.516 | 81.182 | 95.516 |
| FP16 | 81.184 | 95.514 | 81.182 | 95.516 | 81.184 | 95.524 |
| [4,4] | 80.65 | 95.31 | 81.144 | 95.518 | 80.812 | 95.288 |
| [8,4] | 81.108 | 95.49 | 81.162 | 95.488 | 80.782 | 95.232 |
| [8,8] | 81.18 | 95.536 | 81.176 | 95.526 | 81.180 | 95.520 |
| [16,4] | 81.104 | 95.488 | 81.164 | 95.496 | 80.770 | 95.226 |

Table 16: LLaMA and DiT validation on various qunatized precision with each nonlinear operation approximation

| BW | LLaMA | | | DiT | | | | |
|---|---|---|---|---|---|---|---|---|
| | PPL | | | HPSv2 | | | | |
| | SiLU | Softmax | RMSNorm | GELU | SiLU | Softmax | LayerNorm | RMSNorm |
| BL | 7.814 | 7.814 | 7.814 | 0.2724 | 0.2724 | 0.2724 | 0.2724 | 0.2724 |
| FP32 | 7.817 | 7.814 | 7.814 | 0.2728 | 0.2711 | 0.2725 | 0.2724 | 0.2724 |
| [4,4] | 8.013 | 7.814 | 7.867 | 0.2575 | 0.2072 | 0.2716 | 0.2722 | 0.2728 |
| [8,4] | 7.817 | 7.814 | 7.860 | 0.2732 | 0.2062 | 0.2719 | 0.2724 | 0.2729 |
| [8,8] | 7.817 | 7.814 | 7.814 | 0.2728 | 0.2723 | 0.2724 | 0.2727 | 0.2727 |
| [16,4] | 7.816 | 7.814 | 7.862 | 0.2733 | 0.2062 | 0.2718 | 0.2724 | 0.2729 |

