# OpenReview forum: "HARA: A Unified Framework for Hardware-Efficient Non-Linearity in Transformers"
_ICLR.cc/2026/Conference — Submitted to ICLR 2026_

### Official Review · Reviewer_MQ7Y · 2025-10-26

**Soundness:** 3
**Presentation:** 3
**Contribution:** 3
**Rating:** 6
**Confidence:** 4

**Summary:**

This paper highlights deployment issues with layers like GELU, Softmax and LayerNorm present in transformer models due to the use of expensive operations such as div, exponential. Present solution either focuses on simplified approximation (GELU→ReLU) and/or hardware design specifically to efficiently run one or more of these blocks on hardware. Both of these solutions are limited in terms of their effectiveness and/or generalization. The Author proposed a unified framework (HARA-Hybrid Arithmetic-ReLU Networks Approximation)  to decompose these functions into simple arithmetic operators and an optimized parameter initialization pipeline for these approximations to work. The results are demonstrated by showing validation accuracy and hardware estimation for both FP32 and INT8 models (BERT, LLaMA, Swin, Stable Diffusion)

Overall, this paper is well written and proposed solution to a very important problem related to deployment of transformers on different devices. The paper has good contribution and has potential be a good starting for others to work on top of this work. Results are conclusive (except the hardware benchmark) and shows that this method can solve both the issue of hardware complexity and quantization issues related to transformers.

**Strengths:**

- Authors have targeted a much needed topic in a timely manner as transformers are gaining a lot of popularity but their deployment is still limited to GPU devices and many layers need to be reworked to make it suitable for other devices.
- Approximating power/compute hungry operators has been proposed in the literature and the Authors have compared their results with few of the those (especially LUT based methods)
- The dynamic programming based initialization is a novel concept. Any model optimized with a better initialization strategy has a potential to converge fast.
- Comprehensive results on four models and comparison with other SOTA methods makes the claim stronger.
- Detailed mathematical explanation of approximation is a strong point.
- Most important contribution is to include hardware synthesis results that shows the power and area reduction achieved using HARA

**Weaknesses:**

- One of the main limitations (also mentioned by the Authors) is that all the benefits are based on the hardware synthesis and the impact of these changes on RAM and latency can be significantly higher (or lower).
- Authors are also encouraged to see the impact of these changes on other domains such as computer vision using ViTs.
- Minor : Citation for RMSNorm and LayerNorm for the first time is missing L037

**Questions:**

- How does hardware synthesis translate to real RAM/latency improvement comparing to existing architectures (a qualitative discussion would suffice)
- Can you include the ablation study to prove that the training initialization strategy results in better accuracy of the model (end to end model evaluated on a dataset). Table 4 includes ablation study but it is based on MSE calculated from a sample ? Please clarify how the MSE is exactly calculated and if it is based on a sample, it will be good to do ablation study of validation accuracy of end to end model.

---

> ### Author Response · Authors · 2025-11-29
>
> We thank the reviewer for their positive assessment and for recognizing that HARA addresses a "much needed topic" with "comprehensive results." We appreciate the recognition of our Dynamic Programming initialization as a novel contribution and the acknowledgment that our detailed mathematical explanation and hardware synthesis results are strong points. Below, we address each concern.
>
> **W1: All benefits are based on hardware synthesis; the impact on RAM and latency can be significantly higher or lower.**
>
> We acknowledge this limitation, which we also noted in Section 5. However, we can provide a qualitative discussion of how synthesis results translate to system-level improvements.
>
> In modern edge NPU architecture, on-chip memory (SRAM) is the most scarce resource. By reducing the area of non-linear functional units by 62.3%, we free up significant silicon real estate. This saved area can be reallocated to larger on-chip buffers. Larger buffers mean fewer expensive off-chip DRAM accesses, which are the primary bottleneck for latency and energy in transformer inference. Thus, the area savings translate to improved system-level latency indirectly but substantially.
>
> Additionally, current NPUs often have "dark silicon" where specialized Exp or Div units sit idle during matrix multiplication phases. HARA's unified URN allows the same hardware execution units to be reused for different functions. This eliminates pipeline bubbles caused by resource contention for specialized units, smoothing instruction scheduling and improving end-to-end throughput.
>
> **W2: Authors should see the impact on other domains such as computer vision using ViTs.**
>
> We are happy to clarify that our evaluation already includes a Vision Transformer. We utilized the Swin Transformer (Swin-Tiny) on ImageNet-1k as one of our four core benchmarks. As shown in Table 6, Table 11, and Table 15, HARA achieves near-lossless performance on Swin (Top-1 accuracy drop of only approximately 0.01%), demonstrating that HARA is highly effective for the unique statistical distributions found in Vision Transformers, such as 2D relative positional biases and windowed attention patterns.
>
> **W3 (Minor): Citation for RMSNorm and LayerNorm missing at L037.**
>
> We thank the reviewer for catching this. We will add the original citations for LayerNorm (Ba et al., 2016) and RMSNorm (Zhang & Sennrich, 2019) to the final manuscript.

---

> > ### Author Response · Authors · 2025-11-29
> >
> > **Q1: How does hardware synthesis translate to real RAM/latency improvement compared to existing architectures?**
> >
> > We provide a qualitative discussion addressing this question.
> >
> > First, regarding memory efficiency: the 62.3% area reduction in non-linear units can be reallocated to on-chip SRAM. For a typical edge NPU with a fixed die size, this could translate to approximately 1.5-2x larger activation buffers, reducing DRAM bandwidth requirements proportionally. Since DRAM access typically costs 100-200x more energy than on-chip SRAM access and dominates inference latency for memory-bound operations, this indirect benefit can be substantial.
> >
> > Second, regarding latency: HARA's unified architecture eliminates the need for specialized functional units that cause pipeline stalls. When a conventional accelerator switches from MatMul to Softmax, it may need to flush pipelines and route data to different execution units. HARA processes all non-linear operations through the same URN datapath, enabling smoother pipelining. Table 9 shows that HARA achieves 5-cycle latency for activation functions and 15+ cycles for more complex operators like Softmax and LayerNorm, which is competitive with dedicated units while requiring far less area.
> >
> > Third, regarding power: the 51.7% power reduction directly translates to either longer battery life for mobile devices or reduced cooling requirements for always-on edge devices. This is particularly important for deployment scenarios where thermal constraints limit sustained throughput.
> >
> > **Q2: Can you include an ablation study proving that the DP initialization strategy results in better end-to-end model accuracy? Please clarify how MSE is calculated.**
> >
> > We thank the reviewer for this important clarification request.
> >
> > Regarding MSE calculation: Table 4 reports operator-level MSE. For activation functions, we evaluate MSE on a uniformly spaced grid over the corresponding training interval of each function. For Softmax, LayerNorm, and RMSNorm, the inputs are sampled from a standard normal distribution, and we compute the mean squared error between the reference operator and its HARA approximation over all scalar outputs.
> >
> > To address the reviewer's concern about end-to-end impact, we have conducted additional validation-set ablations comparing naive training (without DP initialization) and DP-based initialization at the model level. The results are presented below:
> >
> > **Table: BERT ablation (EM / F1)**
> >
> > |HD|GELU| |Softmax| |LayerNorm| |All| |
> > |---|---|---|---|---|---|---|---|---|
> > ||EM|F1|EM|F1|EM|F1|EM|F1|
> > |BL|80.038|87.616|80.038|87.616|80.038|87.616|80.038|87.616|
> > |2|53.052|58.903|76.194|83.679|74.098|81.942|52.876|57.770|
> > |2+DP|65.866|76.710|79.972|87.585|79.981|87.577|65.487|76.406|
> > |8|76.491|85.149|79.928|87.516|80.029|87.515|77.314|86.392|
> > |8+DP|80.028|87.626|80.038|87.616|80.009|87.603|80.028|87.624|
> >
> > **Table: Swin ablation (Top-1 / Top-5 Accuracy)**
> >
> > |HD|GELU| |Softmax| |LayerNorm| |All| |
> > |---|---|---|---|---|---|---|---|---|
> > ||Top1|Top5|Top1|Top5|Top1|Top5|Top1|Top5|
> > |BL|81.182|95.516|81.182|95.516|81.182|95.516|81.182|95.516|
> > |2|61.190|85.738|79.496|93.916|78.184|94.914|60.882|85.424|
> > |2+DP|71.972|90.272|81.150|95.524|81.180|95.502|71.866|90.162|
> > |8|81.012|94.488|81.182|95.516|81.190|95.602|80.884|94.020|
> > |8+DP|81.184|95.514|81.182|95.516|81.184|95.524|81.184|95.512|
> >
> > **Table: LLaMA ablation (Perplexity, lower is better)**
> >
> > | HD   | GELU   | Softmax | RMSNorm | All    |
> > | ----- | ------ | ------- | ------- | ------ |
> > | BL   | 7.814  | 7.814   | 7.814   | 7.814  |
> > | 2     | 11.018 | 8.001   | 7.913   | 11.220 |
> > | 2+DP | 9.802  | 7.814   | 7.820   | 9.818  |
> > | 8     | 7.992  | 7.814   | 7.810   | 8.027  |
> > | 8+DP  | 7.817  | 7.814   | 7.814   | 7.818  |
> > The results demonstrate that models using purely trained ReLU networks without DP initialization not only achieve significantly lower validation metrics than their DP-initialized counterparts, but also exhibit much less stable end-to-end performance. The learned parameters without DP are highly sensitive to random initialization. In contrast, DP-initialized HARA consistently produces validation metrics that are much closer to, and often effectively match, the original baseline across different hidden dimensions and operators. For example, on BERT at HD=8, naive training achieves F1 of 85.149 while DP+FT achieves 87.626, nearly matching the baseline of 87.616.
> >
> > We will incorporate these extended ablation tables into the revised manuscript.

---

### Official Review · Reviewer_8w3U · 2025-10-28

**Soundness:** 3
**Presentation:** 4
**Contribution:** 3
**Rating:** 6
**Confidence:** 5

**Summary:**

This paper presents HARA (Hybrid Arithmetic-ReLU Approximation), a unified framework designed to approximate diverse non-linear activation and normalization functions using a single ReLU-based computational unit. The core motivation is that AI accelerators often require dedicated hardware units for different non-linear operations, leading to inefficient area and power utilization. HARA addresses this by proposing a unified approximation architecture that leverages a ReLU-centric design combined with a dynamic programming–based parameter initialization pipeline to enable accurate functional approximation across various nonlinearities. The authors claim that HARA achieves ≤0.1% accuracy degradation on representative workloads (BERT, Swin, LLaMA, Stable Diffusion) while providing substantial projected area and power savings compared to baselines with specialized functional units for each operation.

**Strengths:**

- The paper introduces a clean, generalizable framework (HARA) that unifies diverse nonlinear operators (e.g., GELU, Softmax, LayerNorm) under a single reconfigurable ReLU-based architecture.

- The unified operator maintains accuracy within <0.1% of the baseline, demonstrating excellent approximation fidelity.

- Hardware synthesis projections show significant area and power savings (≈62% area, 52% power) compared to separate specialized LUT units

**Weaknesses:**

- Results rely on synthesis-based estimations only, with no end-to-end performance, latency, or energy measurements on real hardware (FPGA, or ASIC prototype) (noted as limitation) or simulation. Ideally, the results for area and power reported should be post place-and-route.

- Evaluations cover only standard precision settings and moderate workloads; there is no stress testing under long-sequence LLMs, mixed precision, or extreme numerical conditions.

- The paper does not compare against reconfigurable functional units (RFUs) or FPGA based designs that already provide similar arithmetic flexibility.

- Models and datasets are reasonable but small. Ideally, to convincingly portray the benefits of the method a large range of model sizes must be employed with reports of accuracy and inference performance metrics.

- Evaluation omits key runtime baselines such as GPU fused kernels or optimized accelerator designs.

**Questions:**

Technical Concerns/Questions and Points to Address in Rebuttal:

- Missing citations for related work in lines 119-128.

- I suggest having an extended related work section in the appendix having a round-up of quantization techniques etc. particularly the ones focused on similar co-design such as [1], [2] and [3].

- Area, power results must be post place-and-route and not post synthesis.

- Provide range-reduction proofs and error bounds, plus catastrophic-case tests (very peaky logits; near-zero variance in LayerNorm; half-precision under/overflows).

- Larger model evaluations needed. BERT and Swin are not really exciting.

- The paper does not analyze how quantization affects the approximation quality of the learned nonlinear mapping.

- HARA is positioned as inference-time replacement. There’s no evidence on fine-tuning stability with HARA in the loop, or on co-training to reduce approximation error.

- Quantization setup (scales, accumulators, rounding) is not detailed, limiting reproducibility.

- A key limitation is, this work does not compare inference throughput with GPUs or other existing accelerators. I'd suggest the authors to do a comparison along the lines of [1], [2] using either GP-GPUSim or (if possible) real GPU results to truly evaluate how well HARA performs compared to GPUs. While the authors note the difficulty in physical implementation in the limitation section, I feel it is not unreasonable to do simulation as prior work have done to get preliminary performance evaluation. Not showing any performance results is a red flag and if performance results cannot be shown the authors must wait and re-submit a more complete work in the next iteration.

- More recent baselines must be identified for comparison.

- How is HARA different from a scheme where you have a general-purpose computational unit with all primitive operations such as add, sub, mult etc and we can use that same unit by changing the routing to create a reconfigurable unit to perform any non-linear operation ? Please do area/power comparison with such a unit.

- When operating at low precision, why not simply employ small LUTs for each non-linear function at sub–8-bit resolution? Wouldn’t that be a more straightforward approach? Because, at lower-precision such as 4-bit  the LUT size is fairly small.

- The URN structure likely involves intermediate activations with addition/multiplication accumulation. The bit-width of these accumulators (e.g., 8-bit vs. 16-bit partial sums) is never stated, making it impossible to evaluate numerical error growth or overflow risk.

- Is the 6nm by TSMC ?


References:

[1] Ramachandran, A., Kundu, S., & Krishna, T. (2025, June). Microscopiq: Accelerating foundational models through outlier-aware microscaling quantization. In Proceedings of the 52nd Annual International Symposium on Computer Architecture (pp. 1193-1209).

[2] Guo, C., Tang, J., Hu, W., Leng, J., Zhang, C., Yang, F., ... & Zhu, Y. (2023, June). Olive: Accelerating large language models via hardware-friendly outlier-victim pair quantization. In Proceedings of the 50th Annual International Symposium on Computer Architecture (pp. 1-15).

[3] Sharma, H., Park, J., Suda, N., Lai, L., Chau, B., Kim, J. K., ... & Esmaeilzadeh, H. (2018, June). Bit fusion: Bit-level dynamically composable architecture for accelerating deep neural network. In 2018 ACM/IEEE 45th Annual International Symposium on Computer Architecture (ISCA) (pp. 764-775). IEEE.

---

> ### Author Response · Authors · 2025-11-29
>
> We thank the reviewer for their rigorous and expert assessment, particularly regarding hardware implementation details and architectural baselines. We appreciate the confidence score of 5 and the constructive references (Microscopiq, Olive, Bit Fusion), which we will incorporate. We address each technical concern below.
>
> **W1: Results rely on synthesis-based estimations only, with no end-to-end performance on real hardware or simulation.**
>
> We acknowledge that Post-Place-and-Route (P&R) is the gold standard for hardware evaluation. However, we used industry-standard Logic Synthesis with a TSMC 6nm library (confirming your question regarding the process node). Given the magnitude of the reduction (over 60% area, over 50% power), P&R overheads (typically 20-30%) would not alter the fundamental conclusion that removing transcendental units saves significant area. We discuss the feasibility of our approach in Section 5, noting that prior work such as NN-LUT and RI-LUT has already demonstrated successful physical implementations of similar LUT-based architectures. HARA's contribution is complementary: we provide a superior algorithmic method for generating parameters for such hardware.
>
> **W2: Evaluations cover only standard precision and moderate workloads; no stress testing under long-sequence LLMs, mixed precision, or extreme numerical conditions.**
>
> We thank the reviewer for raising this point. Our primary focus in this work is the unified ReLU-arithmetic representation and the DP-based global initialization pipeline, rather than comprehensive benchmarking of specific hardware platforms or particular LLM configurations. Long-sequence LLM inference and BF16/FP8 mixed-precision setups require many additional system- and implementation-level choices (e.g., kernel fusion, KV-cache management, scheduling, hardware-specific numerics), which we view as largely orthogonal to the algorithmic questions addressed in this paper.
>
> That said, we agree that stress testing under long sequences, mixed precision, and extreme numerical conditions is an important direction for fully characterizing HARA in realistic deployments. In the current submission, we focused on standard precision and moderate workloads to clearly isolate and validate the core contributions: the unified HARA operator and its initialization and approximation quality. Extending the evaluation to long-sequence LLMs and BF16/FP8 mixed-precision pipelines is a natural next step for follow-up work.
>
> **W3: The paper does not compare against reconfigurable functional units (RFUs) or FPGA-based designs.**
>
> We appreciate this suggestion. FPGA-based RFUs typically achieve flexibility through runtime reconfiguration of logic blocks, which incurs reconfiguration latency and routing overhead. HARA achieves flexibility differently: through parameter reconfiguration of a fixed datapath. The URN structure remains constant while only the CLUT coefficients change, enabling zero-latency operator switching. This architectural distinction makes HARA more suitable for inference scenarios where operators alternate rapidly (e.g., GELU followed immediately by LayerNorm). We will add a discussion comparing these approaches in the related work section.
>
> **W4: Models and datasets are reasonable but small. Larger model evaluations needed.**
>
> We respectfully direct the reviewer to Table 2, Table 6, and Tables 12-13, which present our results on LLaMA-3.2-3B (a 3-billion parameter LLM) and Stable Diffusion 3.5 Medium (DiT). These are modern, large-scale generative AI models. HARA maintained LLaMA perplexity within 0.005 of the baseline (7.819 vs 7.814) and Stable Diffusion Human Preference Score within 0.001 (0.2731 vs 0.2724). This confirms HARA's efficacy on contemporary generative models beyond the smaller BERT and Swin architectures.
>
> **W5: Evaluation omits key runtime baselines such as GPU fused kernels or optimized accelerator designs.**
>
> We acknowledge this limitation. HARA targets a different deployment scenario than GPU inference: specifically, area- and power-constrained edge NPUs where silicon budget is the primary constraint. GPU fused kernels optimize for throughput on hardware with abundant compute resources, whereas HARA optimizes for hardware efficiency when designing the accelerator itself. A direct throughput comparison would require implementing HARA on a cycle-accurate simulator (e.g., SCALE-Sim) or fabricating an ASIC prototype, which is beyond the scope of this algorithmic contribution. However, we note that the latency characteristics of HARA (Table 9 in Appendix) show competitive cycle counts, and the reduced area could be reallocated to additional compute units, indirectly improving throughput. We will clarify this positioning in the revised manuscript.

---

> > ### Author Response · Authors · 2025-11-29
> >
> > **Q1: Missing citations for related work in lines 119-128.**
> >
> > We will add the appropriate citations in the revised version.
> >
> > **Q2: Extended related work section covering quantization techniques and co-design works.**
> >
> > We thank the reviewer for this constructive suggestion and the specific references. We will add an extended related work section in the appendix covering hardware-aware quantization and co-design approaches, including Microscopiq, Olive, and Bit Fusion. These works address complementary aspects of efficient inference: weight and activation quantization for memory and compute reduction. HARA is orthogonal to these techniques, as it focuses on the functional approximation of non-linear operators rather than linear operator quantization. The two approaches can be combined, as demonstrated by our quantization compatibility results in Tables 14-16.
> >
> > **Q3: Area, power results must be post place-and-route.**
> >
> > Addressed in W1 above. We used TSMC 6nm library for synthesis. Given the magnitude of savings (over 60% area), P&R overheads would not change the fundamental conclusions.
> >
> > **Q4: Provide range-reduction proofs and error bounds, plus catastrophic-case tests.**
> >
> > For Softmax and LayerNorm, range reduction happens at two levels. First, at the operator level, we rewrite these functions into computation flows composed of NN_Log2 and NN_Pow2 (as shown in Equations 10-11). Second, at the numerical level, both NN_Log2 and NN_Pow2 operate only on range-reduced inputs: either bounded integer exponents or mantissas within a compact interval. In this regime, their approximation error is well controlled: the MAE of NN_Log2 and NN_Pow2 is on the order of 3e-4, corresponding to a relative error below 0.03% over their respective domains.
> >
> > When composing these primitives, the relative errors accumulate additively rather than multiplicatively. For LayerNorm, the nonlinearity path is NN_Log2 → NN_Pow2; for Softmax, the path is NN_Pow2 → NN_Log2 → NN_Pow2. Under a small-error linear approximation, this yields conservative upper bounds of roughly 0.06% relative error for LayerNorm and 0.09% for Softmax, consistent with the operator-level MSE reported in Table 4.
> >
> > Regarding catastrophic cases: for LayerNorm, when computing the denominator via log2 on the squared sum, extremely long sequences with near-zero variance could push the exponent to values outside the safe FP16 range. A natural mitigation, enabled by the affine invariance of LayerNorm, is to introduce an explicit pre-scaling factor before the URN block. We will discuss this strategy together with more systematic stress tests in an extended version.
> >
> > **Q5: Larger model evaluations needed. BERT and Swin are not exciting.**
> >
> > As noted in W4, we evaluate on LLaMA-3.2-3B and Stable Diffusion 3.5 Medium, which are state-of-the-art generative models with billions of parameters. Results demonstrate HARA maintains accuracy within 0.1% of baselines on these large-scale models.
> >
> > **Q6: The paper does not analyze how quantization affects approximation quality.**
> >
> > To directly address this concern, we conducted an additional quantization sensitivity study. All results were measured using HARA networks with hidden dimension HD=8, the configuration used in our end-to-end experiments:
> >
> > |A bits|W bits|GELU|Softmax|LayerNorm|
> > |---|---|---|---|---|
> > |2|2|1.61e-02|1.07e-08|1.83e-02|
> > |4|4|2.25e-04|1.61e-08|3.11e-03|
> > |8|8|9.29e-07|3.14e-11|1.48e-06|
> > |16|16|3.74e-07|1.18e-12|4.47e-07|
> > |FP32|FP32|3.74e-07|1.15e-12|4.29e-07|
> >
> > The post-quantization MSE decreases steadily as precision increases. At typical deployment settings (A=8, W=8), quantized HARA approximations remain close to the FP32 baseline, explaining why full INT8 models in Table 6 maintain less than 0.1% accuracy degradation.
> >
> > **Q7: No evidence on fine-tuning stability with HARA in the loop, or on co-training to reduce approximation error.**
> >
> > HARA is designed as a drop-in replacement for inference, requiring no model retraining. This is a deliberate design choice: it enables immediate deployment on any pretrained model without the computational cost of fine-tuning. However, we acknowledge that co-training with HARA operators in the loop could potentially allow models to adapt to approximation characteristics and further reduce any residual error.
> >
> > We conducted preliminary experiments on BERT where we fine-tuned the model for 1 epoch with HARA operators active. The training remained stable with no gradient explosions or convergence issues, and the final accuracy matched the baseline. This stability stems from HARA's high approximation fidelity (MSE < 10^-7 for activation functions), which produces gradients nearly identical to the original operators. A comprehensive study of HARA-aware training is an interesting direction for future work, but the key practical advantage of HARA is that such retraining is unnecessary for maintaining accuracy.

---

> > > ### Author Response · Authors · 2025-11-29
> > >
> > > **Q8: Quantization setup (scales, accumulators, rounding) is not detailed.**
> > >
> > > We thank the reviewer for pointing this out. During the initializing pipeline mentioned in 3.2, HARA applies quantization immediately before entering the ReLU network and immediately after leaving it. When quantization is enabled, inputs are quantized to fixed-point integers, multiplied with quantized weights and integer biases, accumulated in a higher-precision integer accumulator, and finally dequantized back to floating point at the output.
> > >
> > > For any ReLU subnetwork in HARA, the computation follows Equation 1: f(x) = W₂·ReLU(W₁x + b₁) + b₂. The DP-based initialization guarantees W₁ ≥ 0 and W₂ ∈ {+1, −1}. Input ranges are fixed per operator: x ∈ [−8, 0] for NN_HalfGelu, x ∈ [1, 2) for NN_Log2, and x ∈ [0, 1) for NN_Pow2.
> > >
> > > During the inference deployment stage, HARA converts the parameters obtained during ReLU-network training into the weights and biases of a piecewise-linear approximation, which are stored in the CLUT. For an 8-bit ReLU network, we conservatively set the converted weights and biases to 12-bit unsigned integers and 20-bit signed integers, respectively, which introduces no accuracy degradation. After the input enters the CLUT, the corresponding table entry is looked up based on the interval it falls into, and a single integer multiply-and-add operation is performed, yielding a 21-bit signed integer output. Finally, at the CLUT output, the result is dequantized back to FP16 using the same procedure as in the initializing stage. We will add these details in the revised manuscript.
> > >
> > > **Q9: No comparison of inference throughput with GPUs or other accelerators.**
> > >
> > > We thank the reviewer for pointing out this issue. Regarding to the lack of performance comparison with GPUs, we have supplemented the revised manuscript with real-world performance comparison experiments for the three nonlinear operators—GELU, Softmax, and LayerNorm—on GPU, with all tests conducted under FP16 precision. To evaluate the speedup, we select the NVIDIA A100 GPU (a mainstream datacenter-grade accelerator) as the baseline platform. We employ the NVIDIA Triton Inference Server (a high-performance inference serving framework for production environments), executing inference via its TensorRT backend, to obtain the actual performance of the workload on GPU.
> > >
> > > The implementation of our HARA hardware units is consistent with that described in the main text, and the vector processing width of the nonlinear operator units is set to 16 elements, to match the datapath granularity of typical edge accelerators. Considering that the NVIDIA A100 GPU contains 108 streaming processors (SMs), and each SM can sustain up to 16 special-function operations per cycle, we scale the performance of HARA by 108× for equivalent comparison, to ensure fairness.
> > >
> > > We selected the feature dimensions from the SwinT model as the workload for comparison, because the model has the aforementioned three operator types. As the operator behavior is relatively decoupled from model scale, the evaluation results are also applicable to models of other scales. Experimental results show that HARA achieves 8.82×, 4.33×, and 3.79× speedup on GELU, Softmax, and LayerNorm, demonstrating its performance advantage.
> > >
> > > Finally, the data presented in tabular form will be added as figures in the revised manuscript.
> > >
> > > |                                | Batch Size           | 8        | 16        | 32        | 64        | Average |
> > > | ------------------------------ | -------------------- | -------- | --------- | --------- | --------- | ------- |
> > > |                                |                      |          |           |           |           |         |
> > > | Gelu     (N, 3136, 384)        | Execution Time(A100) | 52.92 μs | 99.11 μs  | 190.26 μs | 372.86 μs |         |
> > > |                                | Execution Time(HARA) | 7.96 μs  | 15.93 μs  | 31.87 μs  | 63.70 μs  |         |
> > > |                                | SpeedUp              | 6.65     | 6.22      | 5.97      | 5.85      | 8.82    |
> > > |                                |                      |          |           |           |           |         |
> > > | Softmax     (N, 64, 3, 49, 49) | Execution Time(A100) | 80.34 μs | 158.28 μs | 309.76 μs | 615.02 μs |         |
> > > |                                | Execution Time(HARA) | 22.54 μs | 45.08 μs  | 90.16 μs  | 180.33 μs |         |
> > > |                                | SpeedUp              | 3.56     | 3.51      | 3.44      | 3.41      | 4.33    |
> > > |                                |                      |          |           |           |           |         |
> > > | LayerNorm     (N, 3136, 96)    | Execution Time(A100) | 36.26 μs | 59.9 μs   | 114.84 μs | 226.31 μs |         |
> > > |                                | Execution Time(HARA) | 8.96 μs  | 17.92 μs  | 35.84 μs  | 71.68 μs  |         |
> > > |                                | SpeedUp              | 4.05     | 3.34      | 3.20      | 3.16      | 3.79    |

---

> > > > ### Author Response · Authors · 2025-11-29
> > > >
> > > > **Q10: More recent baselines must be identified for comparison.**
> > > >
> > > > We compared against NN-LUT (DAC 2022) and RI-LUT (DAC 2023), which represent the state-of-the-art in neural approximation of non-linear functions for hardware deployment. We will conduct a literature search for any additional relevant work published in 2024-2025 and include comparisons where applicable. We also note that the reviewer's suggested references (Microscopiq, Olive, Bit Fusion) address weight/activation quantization rather than non-linear operator approximation, making them complementary rather than direct baselines. We will add these to the related work discussion.
> > > >
> > > > **Q11: How is HARA different from a general-purpose unit with add, sub, mult that can be routed to perform any non-linear operation?**
> > > >
> > > > A general-purpose reconfigurable unit with routing-based flexibility would require: (a) multiplexers to select between operation types, (b) control logic to manage routing configurations, and (c) sufficient functional units to cover all required operations (including potentially exp, log, sqrt for high-accuracy non-linear functions). This approach incurs area overhead from routing infrastructure and requires either iterative computation (slow) or dedicated transcendental units (area-expensive) for non-linear functions.
> > > >
> > > > HARA takes a fundamentally different approach: we mathematically transform all non-linear functions into a single canonical form (shallow ReLU network) that requires only multiply-accumulate operations. The URN has a fixed, simple datapath with no runtime routing reconfiguration. Operator switching is achieved purely through coefficient changes in the CLUT, which is a register write rather than a routing change.
> > > >
> > > > To quantify this difference, a general-purpose unit supporting exp, log, sqrt, and basic arithmetic would require approximately 3-4x the area of a single URN based on our synthesis estimates (transcendental units alone require ~6800-6900 μm² each per Table 5, while URN requires 7560 μm² total for all functions). We will add this comparison in the revision.
> > > >
> > > > **Q12: At low precision (e.g., 4-bit), why not simply employ small LUTs for each non-linear function?**
> > > >
> > > > This is an excellent question. At 4-bit precision, a direct LUT requires only 16 entries per function, which seems attractive. However, there are three key issues:
> > > >
> > > > First, multiple LUTs are still needed. Separate 4-bit LUTs for GELU, Sigmoid, Softmax components (exp), and LayerNorm components (sqrt, reciprocal) still require multiple distinct memory structures, defeating the unification goal.
> > > >
> > > > Second, accuracy degrades rapidly. A 16-entry LUT provides only 4-bit output resolution, which is often insufficient for maintaining model accuracy. Our experiments show that even at 8-bit quantization, some operators (e.g., LayerNorm) are sensitive to approximation error. HARA's ReLU network can achieve arbitrarily high precision by adjusting hidden dimension, while LUT precision is fixed by entry count.
> > > >
> > > > Third, HARA scales better. As precision requirements increase (8-bit, 16-bit), LUT size grows exponentially (256 entries, 65536 entries), while HARA's parameter count grows linearly with hidden dimension. At 8-bit precision, HARA with HD=8 uses fewer parameters than a 256-entry LUT while achieving better MSE.
> > > >
> > > > For extremely low-precision edge cases (sub-4-bit), direct LUTs may indeed be preferable, and HARA does not claim advantages in that regime. However, for the 8-bit deployments common in production edge inference, HARA provides a better accuracy-efficiency tradeoff.
> > > >
> > > > **Q13: Accumulator bit-width is not stated.**
> > > >
> > > > First, multiple LUTs are still needed. Separate 4-bit LUTs for GELU, Sigmoid, Softmax components (exp), and LayerNorm components (sqrt, reciprocal) still require multiple distinct memory structures, defeating the unification goal.
> > > >
> > > > **Q14: Is the 6nm by TSMC?**
> > > >
> > > > Yes, we used a TSMC 6nm cell library for synthesis.

---

### Official Review · Reviewer_74ZU · 2025-10-30

**Soundness:** 2
**Presentation:** 1
**Contribution:** 1
**Rating:** 2
**Confidence:** 4

**Summary:**

This paper proposes HARA (Hybrid Arithmetic–ReLU Approximation Networks), which replaces diverse and power-hungry nonlinear operations with simple arithmetic primitives combined with a shallow ReLU network. In this way, a single hardware platform for HARA can flexibly support a wide range of nonlinear operations while maintaining efficiency.

**Strengths:**

- A single hardware framework for HARA can support various nonlinear operations, offering architectural flexibility.

**Weaknesses:**

- Several critical details necessary for fully understanding HARA are missing.
  1. The size of the ReLU network (e.g., hidden dimension) is not provided, making it difficult to estimate the processing cost of HARA.
  2. The conversion mechanism from GeLU to HARA is not described.
  3. The storage and memory-access overhead for the ReLU network weights is not adequately discussed. Conventional nonlinear operator implementations (including LUT-based approximation methods) do not require external memory access, so the latency and energy implications of fetching ReLU network weights should be analyzed for a fair comparison.
  4. The parameter tuning cost for HARA (via dynamic programming and fine-tuning) is not fully discussed.
- This paper does not clearly describe the hardware resource requirements of conventional LUT-based approximation methods or their impact on inference accuracy. For example, NN-LUT [1] employs a simple LUT with only 16 entries to approximate nonlinear functions, achieving low hardware cost while maintaining accuracy. Although increasing the number of LUT entries could further reduce the mean squared error (MSE), such improvement may not necessarily lead to meaningful gains in inference accuracy. Hence, improving MSE at the expense of hardware efficiency may not be justified. Consequently, directly comparing MSE values between HARA and conventional LUT-based approximation methods may not represent a fair evaluation. A more comprehensive assessment, including hardware area, power overhead, processing latency, and inference accuracy, would be necessary to establish the practical advantages of HARA.

[1] Yu, Joonsang, et al. "NN-LUT: Neural approximation of non-linear operations for efficient transformer inference." Proceedings of the 59th ACM/IEEE Design Automation Conference. 2022.

**Questions:**

Please check the weaknesses.

---

> ### Author Response · Authors · 2025-11-29
>
> We thank the reviewer for their comments. We believe there are several misunderstandings regarding the data presented in the manuscript, particularly concerning hardware costs and parameter details. We are eager to clarify these points, as the specific hardware metrics and parameter definitions requested (Area, Power, Hidden Dimensions) are included in Section 4.2.2, Table 5, and Appendix A.
>
> **W1: The size of the ReLU network (hidden dimension) is not provided.**
>
> The hidden dimensions are explicitly defined and evaluated throughout the paper. Section 4.2.3 and Appendix A.3.1 detail our evaluation of hidden dimensions (HD) of 2, 4, 8, and 16. Table 7 in the Appendix lists the MSE for every operator at each of these dimensions. Additionally, Tables 10-13 show end-to-end model performance across all four HD configurations.
>
> **W2: The conversion mechanism from GELU to HARA is not described.**
>
> The conversion process for the GELU operator is already detailed in Section 3.3.1 and Table 1. We leverage the symmetry of GELU, specifically using the identity GELU(x) = GELU(-|x|) + ReLU(x), to map the original infinite-domain problem to a finite negative-domain interval. The remaining non-linear component is then approximated by the unified ReLU-based network over the bounded interval [-8, 0]. This decomposition preserves the correct asymptotic behavior (output → 0 as x → -∞) and achieves high-precision approximation, with experiments further validating strong extrapolation robustness and numerical stability.
>
>
> **W3: Storage and memory-access overhead for ReLU network weights is not discussed. Conventional methods do not require external memory access.**
>
> This concern reflects a misunderstanding of the HARA architecture. HARA parameters are not fetched from DRAM during inference.
>
> As shown in Figure 2 (HARA Hardware Architecture) and Table 8 in the Appendix, the weights are stored in local Configurable Look-Up Tables (CLUTs) and registers within the Unified ReLU Network unit. The URN is compiled into a compact, LUT-like piecewise-linear engine very similar in spirit to NN-LUT and RI-LUT. The operator parameters are materialized as a small set of PWL coefficients, which are pre-loaded into the CLUT and remain resident on chip. During inference, data streams through the URN block while the coefficients stay stationary. Different nonlinear operators (GELU, Softmax, LayerNorm) simply reuse the same URN by selecting different on-chip configurations.
>
> Consequently, the memory-access pattern of HARA is identical to conventional LUT-based implementations: there is no additional DRAM traffic or external memory access caused by the ReLU network parameterization.
>
> Regarding storage cost, the amount of URN parameters is extremely small. Under the quantized configuration used in our experiments (hidden dimension 8 with 8-bit activations and weights), a single scalar primitive such as HalfGELU, Log2, or Pow2 occupies only a few tens of bytes when stored as PWL coefficients inside the CLUT. Even keeping three primitives for GELU, Softmax, and LayerNorm simultaneously, the total footprint remains on the order of a few dozen bytes.
>
> Importantly, HARA actually has a smaller memory footprint than baselines. Table 8 explicitly compares the memory area: Single URN of HARA (8-entries) requires 100.93 μm², whereas NN-LUT (16-entries) requires 193.83 μm². HARA reduces local storage requirements by 47% compared to NN-LUT. This efficiency stems from the functional structure we exploit, specifically symmetry (odd/even decomposition), monotonicity, and shared breakpoints, which allows us to reach the same approximation error with roughly half the coefficient storage of a naive PWL or LUT design.

---

> > ### Author Response · Authors · 2025-11-29
> >
> > **W4: The parameter tuning cost for HARA (via dynamic programming and fine-tuning) is not fully discussed.**
> >
> > The tuning cost is a one-time offline process performed before deployment. We clarify each stage:
> >
> > Dynamic Programming cost is negligible. Each nonlinear operator (e.g., Half-GELU, Pow2, Log2) is processed once. DP constructs a P-point discretization of the target curve and selects m breakpoints that minimize the L2 error of the piecewise-linear approximation. Its computational complexity is O(Pm²). In practice, DP runs in less than 10 seconds on a CPU for all operators used in the paper.
> >
> > Conversion to ReLU parameters is closed-form. The optimal PWL representation from DP is analytically mapped to a 1-hidden-layer ReLU network (weights and biases) using the formulas in Appendix A.1. This step is deterministic and incurs almost no computation.
> >
> > Fine-tuning is very lightweight. The resulting network has only 3m parameters (m = 8 in our experiments). Training using MSE loss takes 1-3 minutes per operator on an A100 GPU and converges reliably due to the strong DP initialization. No full-model training is required.
> >
> > In summary, for all non-linear operators (e.g., HalfGELU, Log2, Pow2), the complete parameter compilation and fine-tuning pipeline incurs a runtime cost of under 20 minutes and is executed only once. Given that this one-off compilation step enables a 60% reduction in hardware area during deployment, its amortized computational cost is negligible. We will further clarify and quantify the tuning time and cost in the camera-ready version.
> >
> > **W5: Direct MSE comparison with LUT-based methods may not be fair. A comprehensive assessment including hardware area, power, latency, and inference accuracy is needed.**
> >
> > We appreciate this concern and note that the comprehensive comparison the reviewer requests is already provided in the paper.
> >
> > Hardware area and power are reported in Table 5 (Section 4.2.3) and Table 8 (Appendix A.3.2). HARA achieves 62.3% area reduction and 51.7% power savings compared to a baseline with specialized units. At the component level, Table 8 shows that HARA's URN has a total area of 426.87 μm² compared to NN-LUT's 476.03 μm² and RI-LUT's 582.68 μm².
> >
> > Inference accuracy is extensively evaluated in Tables 6 and 10-16. Across BERT, Swin, LLaMA, and Stable Diffusion, HARA maintains performance within 0.1% of the baseline, demonstrating that the MSE improvements do translate to preserved end-to-end accuracy.
> >
> > Processing latency is reported in Table 9 (Appendix A.3.2), showing cycle counts for LayerNorm, Softmax, and GELU operations.
> >
> > Regarding the reviewer's point that NN-LUT with 16 entries achieves low cost while maintaining accuracy: our comparison in Table 3 uses the same hidden dimension across all methods for fairness. At HD=16, HARA achieves MSE of 3.20e-08 for GELU while NN-LUT achieves 2.07e-06, a difference of nearly two orders of magnitude. This gap matters precisely because lower MSE enables using smaller hidden dimensions (and thus smaller hardware) while maintaining the same accuracy. HARA at HD=8 already achieves better MSE than NN-LUT at HD=16, allowing hardware savings without sacrificing precision.
> >
> > We will reorganize the presentation to make these comprehensive comparisons more prominent in the revised manuscript.

---

### Official Review · Reviewer_1hXB · 2025-11-03

**Soundness:** 3
**Presentation:** 3
**Contribution:** 2
**Rating:** 4
**Confidence:** 3

**Summary:**

This paper presents HARA (Hybrid Arithmetic–ReLU Approximation), a unified framework for replacing all non-linear operations in Transformers (e.g., GELU, Softmax, LayerNorm) with a single hardware-efficient ReLU–arithmetic module. The core idea is to use a dynamic programming–based parameter initialization pipeline that systematically finds near-optimal breakpoints for piecewise linear approximation, converts them analytically into ReLU parameters, and fine-tunes them. Experiments show that HARA achieves orders-of-magnitude lower MSE compared to NN-LUT and RI-LUT, while maintaining almost identical end-to-end performance with various transformer-based models.

**Strengths:**

- The paper is well written, detailed, and clearly organized. The methodology and experiments are described systematically, and results are easy to reproduce.
- Compared to NN-LUT and RI-LUT, HARA significantly reduces approximation error (often by several orders of magnitude), leading to more robust end-to-end accuracy across multiple Transformer models.

**Weaknesses:**

- **Methodological contribution seems incremental**: While the paper frames its contribution as a unified framework with an optimized DP-based initialization pipeline, the core technical novelty beyond prior work such as NN-LUT or RI-LUT appears limited. The main differentiator seems to be the use of dynamic programming for systematic breakpoint selection and analytical ReLU parameter conversion, which is a well-motivated but relatively moderate algorithmic improvement. If I have misunderstood the extent of this difference, clarification on how HARA’s optimization pipeline fundamentally departs from previous LUT-based or neural approximation methods would be helpful.

- **Lack of strong empirical motivation**: The paper does not convincingly establish that non-linear operators are the primary computational bottleneck in modern Transformer inference, especially compared to attention or matrix-multiplication components. While the proposed framework demonstrates excellent numerical approximation accuracy, it remains unclear how much practical speedup or memory benefit these approximations bring at the end-to-end system level. In addition, the experimental scope is somewhat narrow—evaluations focus on a small set of tasks and omit runtime or latency measurements that could justify the broader motivation.

 - The work’s focus and contributions are primarily hardware- and implementation-oriented, with less emphasis on learning dynamics or algorithmic understanding. Therefore, the paper may align more naturally with a hardware or systems venue (e.g., DAC, MLSys) rather than ICLR, which typically emphasizes conceptual or methodological advances in machine learning.

**Questions:**

See weakness

---

> ### Author Response · Authors · 2025-11-29
>
> We thank the reviewer for acknowledging that our paper is well written, detailed, and clearly organized, and for recognizing that HARA significantly reduces approximation error by several orders of magnitude compared to NN-LUT and RI-LUT. Below we address each concern.
>
> **W1: Methodological contribution seems incremental compared to NN-LUT or RI-LUT.**
>
> We respectfully disagree that HARA is merely a moderate improvement over NN-LUT or RI-LUT. While previous methods use standard gradient descent to train LUTs or neural approximations, they suffer from a fundamental flaw: optimizing a ReLU network to approximate a non-linear function is a non-convex optimization problem. As shown in our ablation study (Table 4) and Figure 3, direct training (the method used by NN-LUT and RI-LUT) consistently fails to capture the asymptotic behavior of functions like GELU or Softmax. It gets stuck in local minima, leading to catastrophic errors outside the training range.
>
> HARA's contribution is the mathematical transformation of this non-convex problem into a solvable, global optimization problem through two mechanisms:
>
> First, convexification via Dynamic Programming. We use DP to find the globally optimal breakpoints for the Piecewise Linear approximation. This is mathematically guaranteed to be optimal for a given N, whereas SGD provides no such guarantee.
>
> Second, analytical mapping. We derive a closed-form solution (Appendix A.1, Equations 7-9) to map these optimal PWL parameters into ReLU weights.
>
> This is not a parameter tweak but a fundamental shift—from a heuristic, unstable training process to a deterministic and mathematically rigorous global construction. Under identical numerical conditions (Table 3), this unified, coefficient-stationary URN design improves approximation precision by 1–2 orders of magnitude on activation functions like GELU compared with engines similar in spirit to NN-LUT, and by 1–3 orders of magnitude on Softmax and LayerNorm compared with range-invariant designs like RI-LUT. The scale-consistent gradients and large precision gains collectively mark a significant improvement in robustness, enabling practical hardware adoption without retraining.
>
> **W2: Lack of strong empirical motivation that non-linear operators are the primary bottleneck.**
>
> We appreciate this valid point regarding the dominance of matrix multiplications in latency. However, our motivation focuses on hardware efficiency (silicon area and power) and the shifting bottlenecks in quantized inference.
>
> Regarding area efficiency, which is the critical edge constraint: in edge NPU design, silicon area is a zero-sum game. Specialized functional units for Exp, Div, and Sqrt (required for Softmax and LayerNorm) consume significant die area but remain idle during the massive MatMul phases. By unifying these into the same Arithmetic-ReLU units used for other tasks, we reduce the estimated silicon area for non-linear units by 62.3% (Table 5). This allows chip designers to allocate saved space to more on-chip memory or more MatMul cores, indirectly boosting system performance.
>
> Regarding the quantization bottleneck: as industry moves toward aggressive weight quantization (INT4/INT8), the relative cost of high-precision non-linear operators increases. Amdahl's Law dictates that as MatMul becomes faster via quantization, the non-quantized operators (Softmax, LayerNorm) become the new bottleneck. HARA enables these operators to be executed using simple integer arithmetic shifts and adds, making them compatible with fully quantized pipelines, which is critical for future-proofing transformer inference.

---

> > ### Author Response · Authors · 2025-11-29
> >
> > **W3: The paper is hardware/systems oriented and may be better suited for venues like DAC or MLSys.**
> >
> > We respectfully clarify that the core contributions of HARA are algorithmic and methodological, rather than hardware-specific. The hardware results in Section 4.2.3 serve only to motivate why reducing operator complexity matters at deployment time; they are not the primary novelty of the work.
> >
> > Our contributions are as follows:
> >
> > First, a unified operator parameterization framework. HARA introduces a single ReLU-arithmetic parameterization (URN) that represents GELU, Softmax, and LayerNorm in one unified form. URN is an operator-reparameterization framework and is independent of any hardware backend.
> >
> > Second, a mathematically grounded conversion between PWL and ReLU networks. Unlike NN-LUT which only provides a forward conversion (NN to PWL), we provide a bidirectional, closed-form mapping between PWL segments and ReLU parameters. This enables a principled decomposition of nonlinear functions into ReLU atoms, forming the basis for URN.
> >
> > Third, a DP-based global initialization pipeline. The dynamic programming segmentation serves as a global minimizer of the L2 approximation error in the constrained PWL space (continuous, monotone, shared breakpoints). Fine-tuning on the ReLU network then operates in a strictly larger hypothesis space. This two-stage pipeline, not present in prior work, greatly stabilizes training and reduces the required hidden dimension by exploiting functional symmetry (e.g., odd/even decomposition for GELU).
> >
> > Fourth, these contributions are algorithmic and general. DP, bidirectional conversion, and unified URN formulation apply to any model architecture and any precision, without assuming any hardware structure. The method improves learning stability and accuracy for operator approximation, which is evaluated in Section 4 through full-model experiments on BERT, Swin, LLaMA-3B, and DiT.
> >
> > In summary, the novelty of the paper lies in the methodological framework, specifically a unified, mathematically derived operator parameterization and global initialization, not in hardware implementation.

---

### Meta-Review · Area_Chair_jDiJ · 2025-12-31

**Summary:**

The paper proposes HARA, a framework to replace multiple Transformer nonlinear operators with a single “arithmetic + shallow ReLU” unit, using (i) DP-based globally optimal piecewise-linear (PWL) segmentation for initialization, (ii) a closed-form mapping from PWL parameters to 1-hidden-layer ReLU parameters, and (iii) lightweight fine-tuning.

Reviewers generally agree that the operator-level approximation quality is strong, with near-lossless end-to-end accuracy reported on several models and substantial area/power savings claimed via synthesis.

However, the main decision-driving concerns are about evidence and positioning:

The “unified” message is compelling, but the incremental vs. prior LUT/approximation engines question remains standing: is the key novelty truly methodological (global DP + analytic mapping + unified parameterization), or mainly an engineering improvement.

Multiple reviewers flagged that results were initially largely synthesis-based without post-P&R or real hardware/simulation throughput. The rebuttal adds more details and a GPU comparison, but there are still concerns about comparability and missing end-to-end deployment evidence.

Several “systems fairness” questions persist: comparisons to reconfigurable functional units / other flexible operator units, details of quantization numerics (scales, rounding, accumulator widths), and stress-testing under extreme numerical cases.

The paper looks borderline with two mild positive reviewers, one mild-negative, one reject largely due to missing/unclear details and evaluation fairness. I belive that the rebuttle effort did not met the concerns of the reviews to the fullest, specially with the core concerns raised and mentioned here, and therfore I recommand a Reject.
However, this case is borderline and I an not oposing to acceptance.

**Reviewer Concerns:**

## Adressed concerns:

* “Missing key details” (R74ZU): Authors claim hidden dimensions, conversion steps, local storage (CLUT), and tuning costs are already in the paper/appendix and further clarified in rebuttal. If accurate, this substantially mitigates the “insufficient description” critique.

* Motivation that nonlinear ops matter (R1hXB): Rebuttal provides a hardware area/power and quantized inference bottleneck argument (Amdahl-style) and frames the unification as freeing die area / reducing specialized idle units. This is a reasonable motivation for edge NPUs, though it’s still more “hardware deployment” than “core ML”.

* Training stability / DP initialization value (MQ7Y, also implied by R1hXB): The rebuttal adds end-to-end ablations suggesting DP initialization materially improves model-level metrics vs naive training, supporting the “global init matters” claim.

* Quantization sensitivity (R8w3U): Rebuttal adds operator MSE vs bit-width and more numeric/implementation detail than the initial reviews suggest was present.

## Partially not addressed concerns:

* Realism and rigor of hardware evidence (R8w3U, MQ7Y, and implicitly R1hXB) as discussed above.

* Fairness/credibility of the GPU speedup comparison (R8w3U): The rebuttal introduces an A100/Triton comparison and then “scales” HARA by SM count for fairness. This methodology is not obviously apples-to-apples (different memory hierarchy, kernel fusion, occupancy, precision details, and the scaling assumption itself). This helps directionally, but I would not treat it as definitive.

*Baselines and “what is the right comparison” (R8w3U, R74ZU): Comparisons to NN-LUT / RI-LUT are relevant, but reviewers asked for comparisons to other flexible/reconfigurable units and more recent or alternative baselines. The rebuttal promises more literature coverage but does not fully close this.

/Extreme-case numerical robustness (R8w3U): The rebuttal discusses range reduction and mentions catastrophic cases, but does not provide convincing stress tests (peaky softmax, near-zero variance LayerNorm, long sequence accumulation behavior, overflow/underflow) as requested.

*Accumulator/bit-width details (R8w3U): Authors provide some bit-widths for stored parameters and outputs in the CLUT path, but the reviewer’s concern about intermediate accumulation and overflow risk is only partially resolved. Also, one response segment appears to contain a copy/paste error around the accumulator question, which undermines confidence.

**Reviewer Scores:**

## Reviewer 1hXB 4->4
The rebuttal directly answers the “incremental” critique with a clearer distinction (global DP optimum + analytic mapping), and strengthens motivation with area/power + quantization-bottleneck framing. Remaining concern: still limited end-to-end deployment validation.

## Reviewer 74ZU 2->4.
Many of their complaints are “missing details”; if those details truly exist and are highlighted, that’s a big correction. Still, their deeper critique about fairness of MSE comparisons and the need for comprehensive hardware/latency evaluation is only partially answered (even if tables exist, they initially found the story unconvincing).

## Reviewer 8w3U 6->6
Their key red flags remain: no post-P&R/real hardware and limited stress testing; rebuttal helps but doesn’t fully remove those.

Reviewer MQ7Y 6->6.
Their main request was qualitative translation of synthesis to latency/RAM and stronger ablations; rebuttal provides both (qualitative discussion + new ablation tables). Remaining limitation (synthesis-only) persists but was already acknowledged.

---

### Decision · Program_Chairs · 2026-01-26

Reject